# INVERTIBLE NORMALIZING FLOW NEURAL NETWORKS BY JKO SCHEME

## ABSTRACT

Normalizing flow is a class of deep generative models for efficient sampling and density estimation. In practice, the flow often appears as a chain of invertible neural network blocks. To facilitate training, past works have regularized flow trajectories and designed special network architectures. The current paper develops a neural ODE flow network inspired by the Jordan-Kinderleherer-Otto (JKO) scheme, which allows an efficient *block-wise* training procedure: as the JKO scheme unfolds the dynamic of gradient flow, the proposed model naturally stacks residual network blocks one-by-one and reduces the memory load as well as the difficulty of training deep networks. We also develop an adaptive time-reparametrization of the flow network with a progressive refinement of the trajectory in probability space, which improves the optimization efficiency and model accuracy in practice. On high-dimensional generative tasks for tabular data, JKO-iFlow can process larger data batches and perform competitively as or better than continuous and discrete flow models, using 10X less number of iterations (e.g., batches) and significantly less time per iteration.

## 1 INTRODUCTION

Generative models have been widely studied in statistics and machine learning to infer data-generating distributions and sample from the estimated distributions (Ronquist et al., 2012; Goodfellow et al., 2014; Kingma & Welling, 2014; Johnson & Zhang, 2019). The normalizing flow has recently been a very popular generative framework. In short, a flow-based model learns the data distribution via an invertible mapping $F$ between data density $p_X(X)$, $X \in \mathbb{R}^d$ and the target standard multivariate Gaussian density $p_X(Z)$, $Z \sim \mathcal{N}(0, I_d)$ (Kobyzev et al., 2020). Benefits of the approach include efficient sampling and explicit likelihood computation. To make flow models practically useful, past works have made great efforts to develop flow models that facilitate training (e.g., in terms of loss objectives and computational techniques) and induce smooth trajectories (Dinh et al., 2017; Grathwohl et al., 2019; Onken et al., 2021).

Among flow models, continuous normalizing flow (CNF) transports the data density to that of the target through continuous dynamics (e.g, Neural ODE (Chen et al., 2018)). CNF models have shown promising performance on generative tasks Kobyzev et al. (2020). However, a known computational challenge of CNF models is model regularization, primarily due to the non-uniqueness of the flow transport. To regularize the

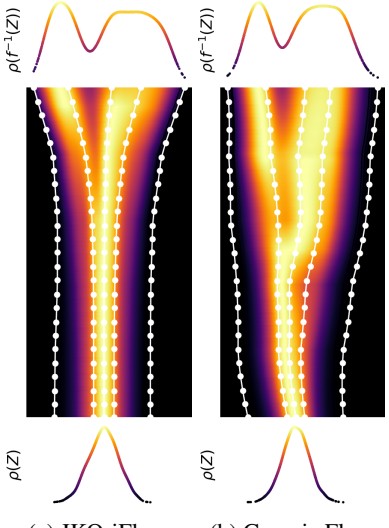

(a) JKO-iFlow    (b) Generic Flow

Figure 1: Comparison of JKO-iFlow (proposed) and other flow models. The JKO scheme approximates the transport of a diffusion process and the ResNet is trained block-wise.

flow model and guarantee invertibility, Behrmann et al. (2019) adopted spectral normalization of block weights that leads to additional computation. Meanwhile, (Liutkus et al., 2019) proposed the sliced-Wasserstein distance, Finlay et al. (2020); Onken et al. (2021) utilized optimal-transport costs, and (Xu et al., 2022) proposed Wasserstein-2 regularization. Although regularization is important to maintain invertibility for general-form flow models and improves performance in practice, merely

using regularization does not resolve non-uniqueness of the flow and there remains variation in the trained flow depending on initialization. Besides unresolved challenges in regularization, there remain several practical difficulties when training such models. In many settings, flows consist of stacked blocks, each of which can be arbitrarily complex. Training such deep models often places high demand on computational resources, numerical accuracy, and memory consumption. In addition, determining the flow depth (e.g., number of blocks) is also unclear.

In this work, we propose JKO-iFlow, a normalizing flow network which unfolds the Wasserstein gradient flow via a neural ODE invertible network, inspired by the JKO-scheme Jordan et al. (1998). The JKO scheme, cf. (5), can be viewed as a proximal step to unfold the Wasserstein gradient flow to minimize the KL divergence (relative entropy) between the current density and the equilibrium. Each block in the flow model implements one step in the JKO-scheme can be trained given the previous blocks. As the JKO scheme pushes forwards the density to approximate the solution of Fokker-Planck equation of a diffusion process with small step-size, the trained flow model induces a smooth trajectory of density evolution, as shown in Figure 1. The theoretical assumption does not incur a restriction in practice when training, whereby one can use larger step sizes coupled with numerical integration techniques. The proposed JKO-iFlow model can be viewed as trained to learn the unique transport map following the Fokker-Planck equation.

Unlike most CNF models where all the residual blocks are initialized together and trained end-to-end, the proposed model allows a block-wise training which reduces memory and computational load. We further introduce time reparametrization with progressive refinement in computing the flow network, where each block corresponds to a point on the density evolution trajectory in the space of probability measures. Algorithmically, one can thus determine the number of blocks adaptively and refine the trajectory determined by existing blocks. Empirically, such procedures yield competitive performance as other CNF models with significantly less computation.

The JKO Flow approach proposed in this work also suggests a potential constructive approximation analysis of deep flow model. Method-wise, the proposed model differs from other recent JKO deep models. We refer to Section 1.1 for more details. In summary, the contribution includes

• We propose a neural ODE model where each residual block computes a JKO step and the training objective can be computed from integrating the ODE on data samples. The network has general form and invertibility can be satisfied due to the regularity of the optimal pushforward map that minimizes the objective in each JKO step.

• We develop an block-wise procedure to train the invertible JKO-iFlow network, which determines the number of blocks adaptively. We also propose a technique to reparametrize and refine an existing JKO-iFlow probability trajectory. Doing so removes unnecessary blocks and increases the overall accuracy.

• Experiment wise, JKO-iFlow greatly reduces memory consumption and the amount of computation, with competitive/better performance as several existing continuous and discrete flow models.

## 1.1 RELATED WORKS

For deep generative models, popular approaches include generative adversarial networks (GAN) (Goodfellow et al., 2014; Gulrajani et al., 2017; Isola et al., 2017) and variational auto-encoder (VAE)(Kingma & Welling, 2014; 2019). Apart from known training difficulties (e.g., mode collapse (Salimans et al., 2016) and posterior collapse (Lucas et al., 2019)), these models do not provide likelihood or inference of data density. The normalizing flow framework (Kobyzev et al., 2020) has been extensively developed, including continuous flow (Grathwohl et al., 2019), Monge-Ampere flow (Zhang et al., 2018), discrete flow (Chen et al., 2019), graph flow (Liu et al., 2019), etc. Efforts have been made to develop novel invertible mapping structures (Dinh et al., 2017; Papamakarios et al., 2017), regularize the flow trajectories (Finlay et al., 2020; Onken et al., 2021), and extend the use to non-Euclidean data (Mathieu & Nickel, 2020; Xu et al., 2022). Despite such efforts, the model and computational challenges of normalizing flow models include regularization and the large model size when using a large number of residual blocks, which cannot be determined a priori, and the associated memory and computational load.

In parallel to continuous normalizing flow which are neural ODE models, neural SDE models become an emerging tool for generative tasks. Diffusion process and Langevin dynamics in deep generative models have been studied in score-based generative models (Song & Ermon, 2019; Ho et al., 2020; Block et al., 2020; Song et al., 2021) under a different setting. Specifically, these models estimate the

score function (i.e., gradient of the log probability density with respect to data) of data distribution via neural network parametrization, which may encounter challenges in learning and sampling of high dimensional data and call for special techniques (Song & Ermon, 2019). The recent work of Song et al. (2021) developed reverse-time SDE sampling for score-based generative models, and adopted the connection to neural ODE to compute the likelihood; using the same idea of backward SDE, Zhang & Chen (2021) proposed joint training of forward and backward neural SDEs. Theoretically, latent diffusion Tzen & Raginsky (2019b;a) was used to analyze neural SDE models. The current work focuses on neural ODE model where the deterministic vector field $\mathbf{f}(x, t)$ is to be learned following a JKO scheme of the Fokker-Planck equation. Rather than neural SDE, our approach involves no sampling of SDE trajectories nor learning of the score function. Our obtained residual network is also invertible, which can not be achieved by the diffusion models above. We experimentally obtain competitive or improved performance against on simulated and high-dimensional tabular data.

JKO-inspired deep models have been studied in several recent works. (Bunne et al., 2022) reformulated the JKO step for minimizing an energy function over convex functions. JKO scheme has also been used to discretize Wasserstein gradient flow to learn a deep generative model in (Alvarez-Melis et al., 2021; Mokrov et al., 2021), which adopted input convex neural networks (ICNN) (Amos et al., 2017). ICNN as a special type of network architecture may have limited expressiveness (Rout et al., 2022; Korotin et al., 2021). In addition to using gradient of ICNN, (Fan et al., 2021) proposed to parametrize the transport in a JKO step by a residual network but identified difficulty in calculating the push-forward distribution. The approach in (Fan et al., 2021) also relies on a variational formulation which requires training an additional network similar to the discriminator in GAN using inner-loops. In contrast, our method trains an invertible neural-ODE flow network which enables the flow from data density to normal and backward as well as the computation of transported density by integrating the divergence of the velocity field along ODE solutions. The objective in JKO step to minimize KL divergence can also be computed directly without any inner-loop training, cf. Section 4.

For the expressiveness of generating deep models, universal approximation properties of deep neural networks for representing probability distributions have been developed in several works. Lee et al. (2017) established approximation by composition of Barron functions (Barron, 1993); Bailey & Telgarsky (2018) developed space-filling approach, which was generalized in Perekrestenko et al. (2020; 2021); Lu & Lu (2020) constructed a deep ReLU network with guaranteed approximation under integral probability metrics, using techniques of empirical measures and optimal transport. These results show that deep neural networks can provably transport one source distribution to a target one with sufficient model capacity under certain regularity conditions of the pair of densities. In our proposed flow model, each residual block is trained to approximate the vector field $\mathbf{f}(x, t)$ that induces the Fokker-Planck equation, cf. Section 3.2. Our model potentially leads to a constructive approximation analysis of neural ODE flow model to generate data density $p_X$.

## 2 PRELIMINARIES

*Normalizing flow.* A normalizing flow can be mathematically expressed via a density evolution equation of $\rho(x, t)$ such that $\rho(x, 0) = p_X$ and as $t$ increases $\rho(x, t)$ approaches $p_Z \sim \mathcal{N}(0, I_d)$ Tabak & Vanden-Eijnden (2010). Given an initial distribtuion $\rho(x, 0)$, such a flow typically is not unique. We consider when the flow is induced by an ODE of $x(t)$ in $\mathbb{R}^d$

$$\dot{x}(t) = \mathbf{f}(x(t), t), \tag{1}$$

where $x(0) \sim p_X$. The marginal density of $x(t)$ is denoted as $p(x, t)$, and it evolves according to the continuity equation (Liouville equation) of (1) written as

$$\partial_t p + \nabla \cdot (p\mathbf{f}) = 0, \quad p(x, 0) = p_X(x). \tag{2}$$

*Ornstein–Uhlenbeck (OU) process.* Consider a Langevin dynamic denoted by the SDE $dX_t = -\nabla V(X_t)dt + \sqrt{2}dW_t$, where $V$ is the potential of the equilibrium density. We focus on the case of normal equilibrium, that is, $V(x) = |x|^2/2$ and then $p_Z \propto e^{-V}$. In this case the process is known as the (multivariate) OU process. Suppose $X_0 \sim p_X$, and let the density of $X_t$ be $\rho(x, t)$ also denoted as $\rho_t(\cdot)$. The Fokker-Planck equation describes the evolution of $\rho_t$ towards the equilibrium $p_Z$ as

$$\partial_t \rho = \nabla \cdot (\rho \nabla V + \nabla \rho), \quad V(x) := |x|^2/2, \quad \rho(x, 0) = p_X(x). \tag{3}$$

Under generic conditions, $\rho_t$ converges to $p_Z$ exponentially fast. For Wasserstein-2 distance and the standard normal $p_Z$, classical argument gives that (take $C = 1$ in Eqn (6) of Bolley et al. (2012))

$$W_2(\rho_t, p_Z) \leq e^{-t} W_2(\rho_0, p_Z), \quad t > 0. \tag{4}$$

*JKO scheme.* The seminal work Jordan et al. (1998) established a time discretization scheme of the solution to (3) by the gradient flow to minimize $\mathrm{KL}(\rho||p_Z)$ under the Wasserstein-2 metric in probability space. Denote by $\mathcal{P}$ the space of all probability densities on $\mathbb{R}^d$ with finite second moment. The JKO scheme at $k$-th step with step size $h > 0$, starting from $\rho^{(0)} = \rho_0 \in \mathcal{P}$, is written as

$$\rho^{(k+1)} = \arg\min_{\rho \in \mathcal{P}} F[\rho] + \frac{1}{2h} W_2^2(\rho^{(k)}, \rho), \quad F[\rho] := \mathrm{KL}(\rho||p_Z). \tag{5}$$

It was proved in Jordan et al. (1998) that as $h \to 0$, $\rho^{(k)}$ converges to the solution $\rho(\cdot, kh)$ of (3) for all $k$, and the convergence $\rho_h(\cdot, t) \to \rho(\cdot, t)$ is strongly in $L^1(\mathbb{R}^d, (0, T))$ for finite $T$, where $\rho_h$ is piece-wise constant interpolated on $(0, T)$ from $\rho^{(k)}$.

## 3 JKO SCHEME BY NEURAL ODE

Given i.i.d. observed data samples $X_i \in \mathbb{R}^d$, $i = 1, \ldots, N$, drawn from some unknown density $p_X$, the goal is to train an invertible neural network to transports the density $p_X$ to an *a priori* specified density $p_Z$ in $\mathbb{R}^d$, where each data sample $X_i$ is mapped to a code $Z_i$. A prototypical choice of $p_Z$ is the standard multivariate Gaussian $\mathcal{N}(0, I_d)$. By a slight abuse of notation, we denote by $p_X$ and $p_Z$ both the distributions and the density functions of data $X$ and code $Z$ respectively.

### 3.1 THE OBJECTIVE OF JKO STEP

We are to specify $\mathbf{f}(x, t)$ in the ODE (1), to be parametrized and learned by a neural ODE, such that the induced density evolution of $p(x, t)$ converges to $p_Z$ as $t$ increases. We start by dividing the time horizon $[0, T]$ into finite subintervals with step size $h$, let $t_k = kh$ and $I_{k+1} := [t_k, t_{k+1})$. Define $p_k(x) := p(x, kh)$, namely the density of $x(t)$ at $t = kh$. The solution of (1) determined by the vector-field $\mathbf{f}(x, t)$ on $t \in I_{k+1}$ (assuming the ODE is well-posed (Sideris, 2013)) gives a one-to-one mapping $T_{k+1}$ on $\mathbb{R}^d$, s.t. $T_{k+1}(x(t_k)) = x(t_{k+1})$ and $T_{k+1}$ transports $p_k$ into $p_{k+1}$, i.e., $(T_k)_{\#} p_{k-1} = p_k$, where we denote by $T_{\#} p$ the push-forward of distribution $p$ by $T$, such that $(T_{\#} p)(\cdot) = p(T^{-1}(\cdot))$.

Suppose we can find $\mathbf{f}(\cdot, t)$ on $I_{k+1}$ such that the corresponding $T_{k+1}$ solves the JKO scheme (5), then with small $h$, $p_k$ approximates the solution to the Fokker-Planck equation 3, which then flows towards $p_Z$. By the Monge formulation of the Wasserstein-2 distance between $p$ and $q$ as $W_2^2(p, q) = \min_{T: T_{\#}p=q} \mathbb{E}_{x \sim p} \|x - T(x)\|^2$, solving for the transported density $p_k$ by (5) is equivalent to solving for the transport $T_{k+1}$ by

$$T_{k+1} = \arg\min_{T: \mathbb{R}^d \to \mathbb{R}^d} F[T] + \frac{1}{2h} \mathbb{E}_{x \sim p_k} \|x - T(x)\|^2, \quad F[T] = \mathrm{KL}(T_{\#} p_k || p_Z). \tag{6}$$

The equivalence between (5) and (6) is proved in Lemma A.1. Furthermore, the following proposition gives that the value of $F[T]$ can be computed from $\mathbf{f}(x, t)$ on $t \in I_{k+1}$ only once $p_k$ is determined by $\mathbf{f}(x, t)$ for $t \le t_k$. The counterpart for convex function based parametrization of $T_k$ was given in Theorem 1 of (Mokrov et al., 2021), where the computation using the change-of-variable differs as we adopt an invertible neural ODE approach here. The proof is left to Appendix A.

**Proposition 3.1.** *Given $p_k$, up to a constant $c$ independent from $\mathbf{f}(x, t)$ on $t \in I_{k+1}$,*

$$\mathrm{KL}(T_{\#} p_k || p_Z) = \mathbb{E}_{x(t_k) \sim p_k} \left( V(x(t_{k+1})) - \int_{t_k}^{t_{k+1}} \nabla \cdot \mathbf{f}(x(s), s) ds \right) + c. \tag{7}$$

By Proposition 3.1, the minimization (6) is equivalent to

$$\min_{\{\mathbf{f}(x, t)\}_{t \in I_{k+1}}} \mathbb{E}_{x(t_k) \sim p_k} \left( V(x(t_{k+1})) - \int_{t_k}^{t_{k+1}} \nabla \cdot \mathbf{f}(x(s), s) ds + \frac{1}{2h} \|x(t_{k+1}) - x(t_k)\|^2 \right), \tag{8}$$

where $x(t_{k+1}) = x(t_k) + \int_{t_k}^{t_{k+1}} \mathbf{f}(x(s), s) ds$. Taking a neural ODE approach, we parametrize $\{\mathbf{f}(x, t)\}_{t \in I_{k+1}}$ as a residual block with parameter $\theta_{k+1}$, and then (8) is reduced to minimizing over $\theta_{k+1}$. This leads to block-wise learning algorithm to be introduced in Section 4.

## 3.2 INFINITESIMAL OPTIMAL $\mathbf{f}(x,t)$

In each JKO step of (8), let $p = p_k$ denote the current density, $q = p_Z$ be the target equilibrium density. In this subsection, we show that the optimal $\mathbf{f}$ in (8) with small $h$ reveals the difference between score functions between target and current densities. Thus minimizing the objective (8) searches for a neural network parametrization of the score function $\nabla \log \rho_t$ without denoising score matching as in diffusion-based models (Ho et al., 2020; Song et al., 2021).

Consider general equilibrium distribution $q$ with a differentiable potential $V$. To analyze the optimal pushforward mapping in the small $h$ limit, we shift the time interval $[kh, (k+1)h]$ to be $[0, h]$ to simplify notation. Then (8) is reduced to

$$\min_{\{\mathbf{f}(x,t)\}_{t \in [0,h)}} \mathbb{E}_{x(0) \sim p} \left( V(x(h)) - \int_0^h \nabla \cdot \mathbf{f}(x(s), s) ds + \frac{1}{2h} \|x(h) - x(0)\|^2 \right), \quad (9)$$

where $x(h) = x(0) + \int_0^h \mathbf{f}(x(s), s) ds$. In the limit of $h \to 0+$, formally, $x(h) - x(0) = h\mathbf{f}(x(0), 0) + O(h^2)$, and suppose $V$ of $q$ is $C^2$, $V(x(h)) = V(x(0)) + h\nabla V(x(0)) \cdot \mathbf{f}(x(0), 0) + O(h^2)$. For any differentiable density $\rho$, the (Stein) score function is defined as $\mathbf{s}_\rho = \nabla \log \rho$, and we have $\nabla V = -\mathbf{s}_q$. Taking the formal expansion of orders of $h$, the objective in (9) is written as

$$\mathbb{E}_{x \sim p} \left( V(x) + h \left( -\mathbf{s}_q(x) \cdot \mathbf{f}(x, 0) - \nabla \cdot \mathbf{f}(x, 0) + \frac{1}{2} \|\mathbf{f}(x, 0)\|^2 \right) + O(h^2) \right). \quad (10)$$

Note that $\mathbb{E}_{x \sim p} V(x)$ is independent of $\mathbf{f}(x, t)$, and the $O(h)$ order term in (10) is over $\mathbf{f}(x, 0)$ only, thus the minimization of the leading term is equivalent to

$$\min_{\mathbf{f}(\cdot) = \mathbf{f}(\cdot, 0)} \mathbb{E}_{x \sim p} \left( -T_q \mathbf{f} + \frac{1}{2} \|\mathbf{f}\|^2 \right), \quad T_q \mathbf{f} := \mathbf{s}_q \cdot \mathbf{f} + \nabla \cdot \mathbf{f}, \quad (11)$$

where $T_q$ is known as the Stein operator (Stein, 1972). The $T_q \mathbf{f}$ in (11) echoes that the derivative of KL divergence with respect to transport map gives Stein operator (Liu & Wang, 2016). The Wasserstein-2 regularization gives an $L^2$ regularization in (11). Let $L^2(p)$ be the $L^2$ space on $(\mathbb{R}^d, p(x)dx)$, and for vector field $\mathbf{v}$ on $\mathbb{R}^d$, $\mathbf{v} \in L^2(p)$ if $\int |\mathbf{v}(x)|^2 p(x) dx < \infty$. One can verify that, when both $\mathbf{s}_p$ and $\mathbf{s}_q$ are in $L^2(p)$, the minimizer of (11) is

$$\mathbf{f}^*(\cdot, 0) = \mathbf{s}_q - \mathbf{s}_p.$$

This shows that the infinitesimal optimal $\mathbf{f}(x, t)$ equals the difference of the score functions of the equilibrium and the current density.

## 3.3 INVERTIBILITY OF FLOW MODEL AND EXPRESSIVENESS

At time $t$ the current density of $x(t)$ is $\rho_t$, the analysis in Section 3.2 implies that the optimal vector field $\mathbf{f}(x, t)$ has the expression as

$$\mathbf{f}(x, t) = \mathbf{s}_q - \mathbf{s}_{\rho_t} = -\nabla V - \nabla \log \rho_t. \quad (12)$$

With this $\mathbf{f}(x, t)$, the Liouville equation (2) coincides with the Fokker-Planck equation (3). This is consistent with that JKO scheme with small $h$ recovers the solution to the Fokker-Planck equation. Under proper regularity condition of $V$ and the initial density $\rho_0$, the r.h.s. of (12) is also regular over space and time. This leads to two consequences, in approximation and in learning: Approximation-wise, the regularity of $\mathbf{f}(x, t)$ allows to construct a $k$-th residual block in the flow network to approximate $\{\mathbf{f}(x, t)\}_{t \in I_k}$ when there is sufficient model capacity, by classical universal approximation theory of shallow networks (Barron, 1993; Yarotsky, 2017). The JKO-iFlow model proposed in this work suggests a constructive proof of the expressiveness of the invertible neural ODE model to generate any sufficiently regular density $p_X$, which we further discuss in the last section.

For learning, when properly trained with sufficient data, the neural ODE vector field $\mathbf{f}(x, t; \theta_k)$ will learn to approximate (12). This can be viewed as inferring the score function of $\rho_t$, and also leads to invertibilty of the trained flow net in theory: Suppose the trained $\mathbf{f}(x, t; \theta_k)$ is close enough to (12), it will also has bounded Lipschitz constant. Then the residual block is invertible as long as the step size $h$ is sufficiently small, e.g. less than $1/L$ where $L$ is the Lipschitz bound of $\mathbf{f}(x, t; \theta_k)$. In practice, we typically use smaller $h$ than needed merely by invertibility (allowed by model budget) so that the flow network can more closely track the Fokker-Planck equation of the diffusion process. The invertibility of the proposed model is numerically verified in experiments (see Table 1).

## 4 TRAINING OF JKO-iFLOW NET

### 4.1 BLOCK-WISE TRAINING

Note that the training of $(k + 1)$-th block in (8) can be conducted once the previous $k$ blocks are trained. Specifically, with finite training data $\{X_i = x_i(0)\}_{i=1}^n$, the expectation $\mathbb{E}_{x(t)\sim p_k}$ in (8) is replaced by the sample average over $\{x_i(kh)\}_{i=1}^n$ which can be computed from the previous $k$ blocks. Note that for each given $x(t) = x(t_k)$, both $x(t_{k+1})$ and the integral of $\nabla \cdot \mathbf{f}$ in (8) can be computed by a numerical neural ODE integrator. Following previous works, we use the Hutchinson trace estimator (Hutchinson, 1989; Grathwohl et al., 2019) to estimate the quantity $\nabla \cdot \mathbf{f}$ in high dimensions. Applying the numerical integrator in computing (8), we denote the resulting $k$-th residual block abstractly as $f_{\theta_k}$ with trainable parameters $\theta_k$.

This leads to a block-wise training of the normalizing flow network, as summarized in Algorithm 1. Regarding input parameters,, we found the generative performance JKO-iFlow may vary depending on starting choices of $t_k$, but a simple choice such as $t_k = k$ often yields reasonably good performance. We discuss further the initial selection of $t_k$ in Appendix C.1. Meanwhile, one can use any suitable termination criterion Ter(k) in line 2 of Algorithm 1. In our experiments, we monitor the per-dimension $W_2$ loss $W_2^2(T_\# p_k, p_k)$ as defined in (6), and terminate training more blocks if the per-dimension loss is below $\epsilon$. Lastly, the heuristic approach in line

---

**Algorithm 1** Block-wise JKO-iFlow training

**Require:** Time stamps $\{t_k\}$, training data, termination criterion Ter and tolerance level $\epsilon$, maximal number of blocks $L_{\max}$.
1: Initialize $k = 1$.
2: **while** Ter(k) $> \epsilon$ and $k \leq L_{\max}$ **do**
3:  Optimize $f_{\theta_k}$ upon minimizing (8) with mini-batch sample approximation, given $\{f_{\theta_i}\}_{i=1}^{k-1}$. Set $k \leftarrow k + 1$.
4: **end while**
5: $L \leftarrow k$. Optimize $f_{\theta_{L+1}}$ using (8) with $h = \infty$. {▷ Free block, no $W_2$ regularization.}

---

5 of training a "free block" (i.e., block without the $W_2$ loss) is to flow the push-forward density $p_L$ closer to $p_Z$, where the former is obtained through the first $L$ blocks and the latter denotes the Gaussian density at equilibrium.

Note that Algorithm 1 significantly reduces memory and computational complexity: only one block is trained when optimizing (8), regardless of flow depth. Therefore, one can use larger data batches and more refined numerical integrator without memory explosion. In addition, one can train each block for a fixed number of epochs using either back-propagation or the NeuralODE integrator (Grathwohl et al., 2019, adjoint method). We found direct back-propagation enables faster training but may also lead to greater numerical errors and memory consumption. Despite greater inaccuracies, we observed similar empirical performances across both methods for JKO-iFlow, possibly due to the block-wise training that accumulates fewer errors than a generic flow model composed of multiple blocks.

### 4.2 IMPROVED COMPUTATION OF TRAJECTORIES IN PROBABILITY SPACE

We adopt two additional computational techniques to facilitate learning of the trajectories in the probability space, represented by the sequence of densities $p_k$, $k = 1, \cdots, K$, associated with the $K$ residual blocks of the proposed normalizing flow network. The two techniques are illustrated in Figure 2. Additional details can be found in Appendix B.

• *Trajectory reparametrization*. We empirically observe fast decay of the movements $W_2^2(T_\# p_k, p_k)$; in other words, initial blocks transport the densities much further than the later ones. This is especially unwanted because in order to train the current block, the flow model needs to transport data through all previous blocks, yet the current block barely contributes to the density transport. Hence, instead of having $t_k := kh$ with fixed increments per block, we *reparametrize* the values of $t_k$ through an adaptive procedure, which is entirely based on the $W_2$ distance at each block and the averaged $W_2$ distance over all blocks.

• *Progressive refinement*. To improve the probability trajectory obtained by the trained residual blocks, we propose a refinement technique that trains additional residual blocks based on time steps $t_k$ obtained after reparametrization. In practice, refinement can be useful when the time increment $t_{k+1} - t_k$ for certain blocks is too large. In those cases, there may exist numerical inaccuracies as the loss (8) is computed over a longer time horizon. More precisely, we increase the number of JKO-steps parametrized by residual blocks, where in practice, we training $C$ additional "intermediate" blocks for density transport between $p_k$ and $p_{k+1}$ at each $k$.

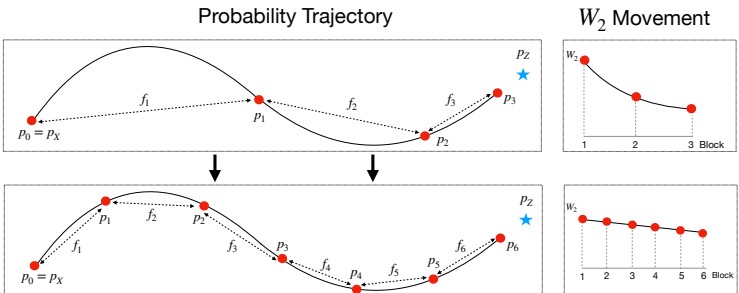

Figure 2: Diagram illustrating trajectory reparametrization and refinement. The top panel shows the original trajectory under three blocks via Algorithm 1. The bottom panel shows the trajectory under six blocks after reparametrization and refinement, which renders the W2 movements more even.

## 5 EXPERIMENT

We first generate based on two-dimensional simulated samples. We then perform unconditional and conditional generation on high-dimensional real tabular data. We also show JKO-iFlow's generative performance on MNIST. Additional details are in Appendix C.

### 5.1 SETUP

**Competing Methods and metrics.** We compare JKO-iFlow with five other models, including four flow-based model and one diffusion model. The first two continuous flow models are FFJORD (Grathwohl et al., 2019) and OT-Flow (Onken et al., 2021). The next two discrete flow models are IResNet (Behrmann et al., 2019) and IGNN (Xu et al., 2022), which replaces the expensive spectral normalization in IResNet with Wasserstein-2 regularization to promote smoothness. The last diffusion model is the score-based generative modeling based on neural stochastic differential equation (Song et al., 2021), which we call it ScoreSDE for comparison. We are primarily interested in two types of criteria. The first is the *computational efficiency* in terms of the number of iterations (e.g., batches that the model uses in training) and training time per iteration. Due to the block-wise training scheme,

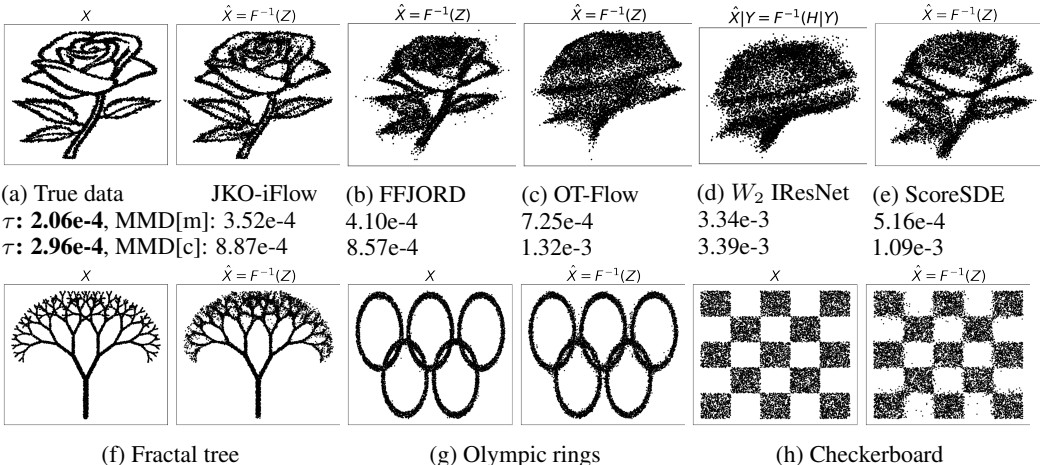

(a) True data     JKO-iFlow     (b) FFJORD     (c) OT-Flow     (d) $W_2$ IResNet     (e) ScoreSDE
$\tau$: **2.06e-4**, MMD[m]: 3.52e-4     4.10e-4     7.25e-4     3.34e-3     5.16e-4
$\tau$: **2.96e-4**, MMD[c]: 8.87e-4     8.57e-4     1.32e-3     3.39e-3     1.09e-3

(f) Fractal tree         (g) Olympic rings         (h) Checkerboard

Figure 3: Two-dimensional simulated datasets. The generated samples $\hat{X}$ by JKO-iFlow in (a) are closer to the true data $X$ than competitors in (b)-(e). Under the more carefully selected bandwidth via the sample-median technique, MMD[m] in (20) by JKO-iFlow is also closer to the threshold $\tau$ in (21) than others. (f)-(h) visualizes generation by JKO-iFlow on more examples.

Table 1: Inversion error $\mathbb{E}_{x\sim p_X}\|T_\theta^{-1}(T_\theta(x)) - x\|_2$ of JKO-iFlow computed from sample average on test data, where $T_\theta$ denotes the transport mapping over all the blocks of the trained flow network.

| POWER | GAS | MINIBOONE | BSD300 | Rose | Fractal tree | Olympic rings | Checkerboard |
|-------|-----|-----------|--------|------|-------------|---------------|--------------|
| 1.48e-5 | 1.58e-6 | 1.09e-6 | 1.53e-5 | 3.30e-6 | 3.58e-5 | 2.24e-6 | 3.07e-5 |

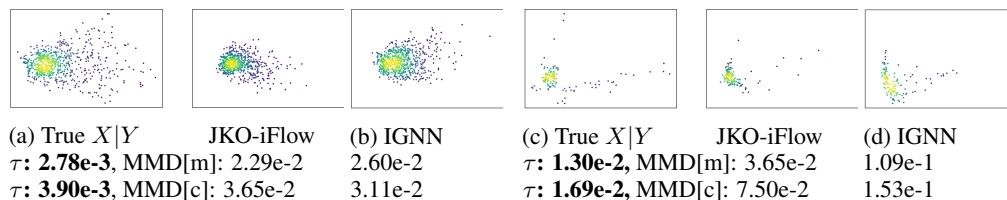

| (a) True $X\|Y$ | JKO-iFlow | (b) IGNN | (c) True $X\|Y$ | JKO-iFlow | (d) IGNN |
|---|---|---|---|---|---|
| $\tau$: **2.78e-3**, MMD[m]: 2.29e-2 | | 2.60e-2 | $\tau$: **1.30e-2**, MMD[m]: 3.65e-2 | | 1.09e-1 |
| $\tau$: **3.90e-3**, MMD[c]: 3.65e-2 | | 3.11e-2 | $\tau$: **1.69e-2**, MMD[c]: 7.50e-2 | | 1.53e-1 |

Figure 4: Conditional graph node feature generation by JKO-iFlow and iGNN. We visualize the conditionally generated samples upon projecting down to the first two principal components determined by true $X|Y$. We visualize generation at two different values of $Y$.

Table 2: Numerical metrics on high-dimensional real datasets. All competitors are trained after 10 times more iterations (i.e., batches), because their performance under the same number of iterations is not comparable to JKO-iFlow. Complete results are shown in Table A.1.

| Data Set | Model | # Param | Test MMD[m] | Test MMD[c] | Data Set | Model | # Param | Test MMD[m] | Test MMD[c] |
|---|---|---|---|---|---|---|---|---|---|
| | | | $\tau$: **1.75e-4** | $\tau$: **2.84e-4** | | | | $\tau$: **4.59e-4** | $\tau$: **6.87e-4** |
| **POWER** $d=6$ | JKO-iFlow | 76K | 8.21e-4 | 1.26e-3 | **MINIBOONE** $d=43$ | JKO-iFlow | 112K | 7.97e-4 | 1.01e-3 |
| | OT-Flow | 76K | 5.69e-4 | 9.62e-4 | | OT-Flow | 112K | 1.23e-3 | 1.01e-3 |
| | FFJORD | 76K | 1.38e-3 | 1.98e-3 | | FFJORD | 112K | 5.47e-3 | 1.04e-3 |
| | $W_2$ IResNet | 304K | 2.76e-3 | 2.72e-3 | | $W_2$ IResNet | 448K | 1.27e-2 | 1.03e-3 |
| | IResNet | 304K | 4.50e-3 | 2.49e-2 | | IResNet | 448K | 2.58e-3 | 1.04e-3 |
| | ScoreSDE | 76K | 1.34e-3 | 6.56e-3 | | ScoreSDE | 112K | 4.29e-3 | 1.03e-3 |
| | | | $\tau$: **1.93e-4** | $\tau$: **2.83e-4** | | | | $\tau$: **1.35e-4** | $\tau$: **9.63e-5** |
| **GAS** $d=8$ | JKO-iFlow | 76K | 5.96e-4 | 1.79e-3 | **BSDS300** $d=63$ | JKO-iFlow | 396K | 4.83e-3 | 3.03e-3 |
| | OT-Flow | 76K | 1.51e-3 | 3.64e-3 | | OT-Flow | 396K | 8.55e-2 | 8.44e-2 |
| | FFJORD | 76K | 3.62e-3 | 6.09e-3 | | $W_2$ IResNet | 990K | 5.52e-1 | 6.88e-1 |
| | $W_2$ IResNet | 304K | 7.14e-3 | 1.50e-2 | | IResNet | 990K | 5.42e-1 | 5.95e-1 |
| | IResNet | 304K | 3.26e-3 | 2.72e-2 | | ScoreSDE | 396K | 5.51e-1 | 6.62e-1 |
| | ScoreSDE | 76K | 1.31e-3 | 1.45e-3 | | | | | |

we measure the number of iterations for JKO-iFlow as the sum of iterations over all blocks. Using this metric allows us to examine performance easily across models, under a fixed-budget framework in terms of batches available to the model. The second is the *maximum mean discrepancy* (MMD) comparison (Gretton et al., 2012; Onken et al., 2021), which is a way of measuring the difference between two distributions based on samples. Additional details for MMD appear in Appendix C.4. We also report negative log-likelihood as an additional metric in Table A.1.

**Conditional Generation.** Due to the increasing need for conditional generation, we also apply JKO-iFlow for conditional generation: generate samples based on the conditional distribution $X|Y$. Most existing conditional generative methods treat $Y$ as an additional input of the generator, leading to potential training difficulties. Instead, we follow the IGNN approach (Xu et al., 2022), which also incurs minimal changes to our training. Additional details are in Appendix C.5.

## 5.2 RESULTS

**Two-dimensional toy data.** Figures 3a—3d compare JKO-iFlow with the competitors on non-conditional generation, where the subcaption indicates MMD values (20) under both bandwidths and the corresponding thresholds $\tau$ in (21). We omit showing IResNet with spectral normalization as it yields similar results as $W_2$ IResNet. The generative quality by JKO-iFlow is the closest to that of the ground truth, and when the MMD bandwidth is more carefully selected via the sample-median technique, JKO-iFlow also yields smaller MMD than others. Meanwhile, Figures 3f–3h shows the satisfactory generative performance by JKO-iFlow on other examples. In Appendix, Figure A.2 compares the performance of JKO-iFlow before and after using the technique described in Section 4.2, where the generative quality by JKO-iFlow improves after several reparametrization moving iterations, and Figure A.4 shows additional unconditional and conditional generation results.

**High-dimensional tabular data**. In terms of conditional graph node feature generation, Figure 4 compares JKO-iFlow with IGNN on the solar dataset introduced in iGNN. The results show that JKO-iFlow yields competitive or clearly better MMD values on the conditional distribution $X|Y$ with the most or second most observations, respectively. Next, Table 2 assesses the performance of JKO-iFlow and competitors on four high-dimensional real datasets, where JKO-iFlow still has the best overall performance. Dataset details are described in Appendix C.3. To ensure a fair comparison, we keep the number of parameters for continuous flow models and the diffusion model the same and properly increase model sizes for discrete flow models. FFJORD results on BSDS300 are omitted due to incomparably longer training time. In terms of results, except on POWER, JKO-iFlow yields smaller or very similar MMD under both bandwidths than all other methods. Table A.1 in Appendix

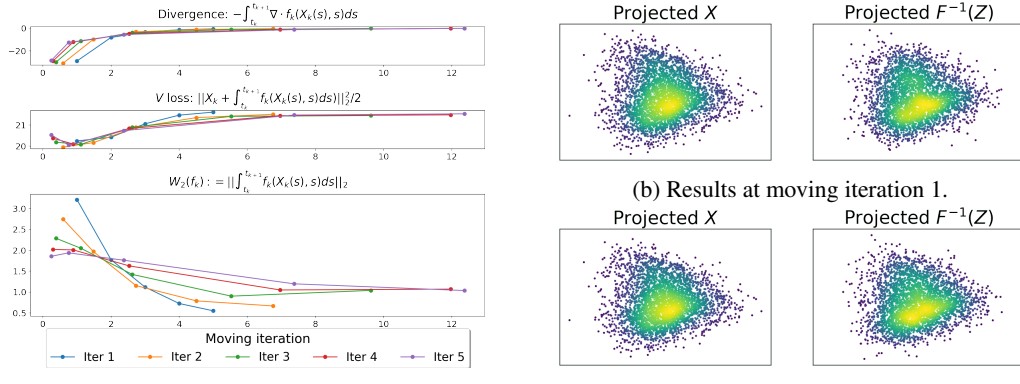

(a) Components of loss (8) over moving iterations.

(b) Results at moving iteration 1.

(c) Results at moving iteration 5.

Figure 5: MINIBOONE, reparametrization moving iterations of JKO-iFlow. We plot different components of the loss objective (8) over $t_k$. In (a), results at moving iteration 5 are obtained by using Algorithm 2 (modified for training flow model) 4 times, and the reparametrization gives more uniform $W_2$ losses after moving iterations. On this example, the generative performance are both good before and after the moving iterations, cf. plots (b) and (c).

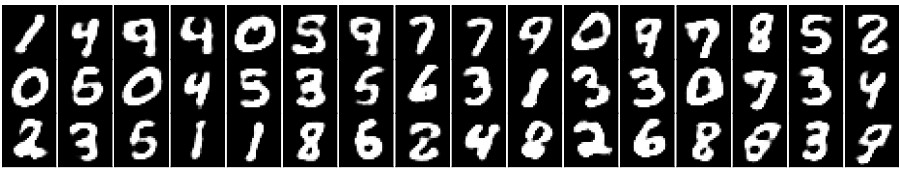

Figure 6: MNIST generation by JKO-iFlow coupled with a pre-trained auto-encoder.

C shows the complete results, including the number of training iterations and test log-likelihood. Overall, we remark that comparisons using MMD[m] (i.e., MMD with bandwidth selected using the sample-median technique) best align with visual comparisons in Figure A.1 of Appendix C.6, so that we suggest MMD[m] as a more reliable metric out of others we used. Furthermore, we illustrate the reparametrization technique on MINIBOONE in Figure 5, where the benefit appears in yielding a flow trajectory with more uniform movement under a competitive generative performance.

**MNIST**. We illustrate the generative quality of JKO-iFlow using an AutoEncoder. Consider a pre-trained encoder $\mathrm{Enc} : \mathbb{R}^{784} \to \mathbb{R}^d$ and decoder $\mathrm{Dec} : \mathbb{R}^d \to \mathbb{R}^{784}$ such that $\mathrm{Dec}(\mathrm{Enc}(X)) \approx X$ for a flattened image $X$. We choose $d = 16$. The encoder (resp. decoder) uses one fully-connected layer followed by the ReLU (resp. Sigmoid) activation. Then, JKO-iFlow is trained on $N$ encoded images $\{\mathrm{Enc}(X_i)\}_{i=1}^N$, and the trained model gives an invertible transport mapping (over all residual blocks) $T_\theta : \mathbb{R}^d \to \mathbb{R}^d$. The images are generated upon sampling noises $Z \sim \mathcal{N}(0, I_d)$ through the backward flow followed by the decoder, namely $\mathrm{Dec}(T_\theta^{-1}(Z))$. The generated images are shown in Figure 6.

## 6 DISCUSSION

The work can be extended in several directions. The application to larger-scale image dataset by adopting convolutional layers will further verify the usefulness of the proposed method. The applications to generative tasks on graph data, by incorporating graph neural network layers in JKO-iFlow model, are also of interest. This also includes conditional generative tasks, of which the first results on toy data are shown in this work. For the methodology, the time-continuity over the parametrization of the residual blocks (as a result of the smoothness of the Fokker-Planck flow) have not been exploited in this work, which may further improve model capacity as well as learning efficiency. Theoretically, the model expressiveness of flow model to generate any regular data distribution can be analyzed based on Section 3.3. To sketch a road-map, a block-wise approximation guarantee of $\mathbf{f}(x, t)$ as in (12) can lead to approximation of the Fokker-Planck flow (3), which pushes forward the density to be $\epsilon$-close to normality in $T = \log(1/\epsilon)$ time, cf. (4). Reversing the time of the ODE then leads to an approximation of the initial density $\rho_0 = p_X$ by flowing backward in time from $T$ to zero. Further analysis under technical assumptions is left to future work.

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

# A PROOFS

## A.1 PROOFS IN SECTION

**Lemma A.1.** *Suppose $p$ and $q$ are two densities on $\mathbb{R}^d$ in $\mathcal{P}$, the following two problems*

$$\min_{\rho \in \mathcal{P}} L_\rho[\rho] = \mathrm{KL}(\rho||q) + \frac{1}{2h}W_2^2(p, \rho), \tag{13}$$

$$\min_{T:\mathbb{R}^d \to \mathbb{R}^d} L_T[T] = \mathrm{KL}(T_\#p||q) + \frac{1}{2h}\mathbb{E}_{x \sim p}\|x - T(x)\|^2, \tag{14}$$

*have the same minimum, and*

*(a) If $T^* : \mathbb{R}^d \to \mathbb{R}^d$ is a minimizer of (14), then $\rho^* = (T^*)_\#p$ is a minimizer of (13).*

*(b) If $\rho^*$ is a minimizer of (13), then the optimal transport from $p$ to $\rho^*$ minimizes (14).*

*Proof of Lemma A.1.* Let the minimum of (14) be $L_T^*$, and that of (13) be $L_\rho^*$.

Proof of (a): Suppose $L_T$ achieves minimum at $T^*$, then $T^*$ is the optimal transport from $p$ to $\rho^* = (T^*)_\#p$ because otherwise $L_T$ can be further improved. By definition of $L_\rho$, we have $L_T^* = L_T[T^*] = L_\rho[\rho^*] \geq L_\rho^*$. We claim that $L_T^* = L_\rho^*$. Otherwise, there is another $\rho'$ such that $L_\rho[\rho'] < L_T^*$. Let $T'$ be the optimal transport from $p$ to $\rho'$, and then $L_T[T'] = L_\rho[\rho'] < L_T^*$, contradicting with that $L_T^*$ is the minimum of $L_T$. This also shows that $L_\rho[\rho^*] = L_T^* = L_\rho^*$, that is, $\rho^*$ is a minimizer of $L_\rho$.

Proof of (b): Suppose $L_\rho$ achieves minimum at $\rho^*$. Let $T^*$ be the OT from $p$ to $\rho^*$, then $\mathbb{E}_{x \sim p}|x - T^*(x)|^2 = W_2(p, \rho^*)^2$, and then $L_T[T^*] = L_\rho[\rho^*] = L_\rho^*$ which equals $L_T^*$ as proved in (a). This shows that $T^*$ is a minimizer of $L_T$. $\qquad\square$

*Proof of Proposition 3.1,* Given $p_k$ being the density of $x(t)$ at $t = kh$, recall that $T$ is the solution map from $x(t)$ to $x(t + h)$. We denote $\rho_t := p_k$, and $\rho_{t+h} := T_\#p_k$. By definition,

$$\mathrm{KL}(T_\#p_k||p_Z) = \mathbb{E}_{x \sim \rho_{t+h}}(\log \rho_{t+h}(x) - \log p_Z(x)). \tag{15}$$

Because $p_Z \propto e^{-V}$, $V(x) = |x|^2/2$, we have $\log p_Z(x) = -V(x) + c_1$ for some constant $c_1$. Thus

$$\mathbb{E}_{x \sim \rho_{t+h}} \log p_Z(x) = \mathbb{E}_{x(t) \sim \rho_t} \log p_Z(x(t + h)) = c_1 - \mathbb{E}_{x(t) \sim \rho_t} V(x(t + h)). \tag{16}$$

To compute the first term in (15), note that

$$\mathbb{E}_{x \sim \rho_{t+h}} \log \rho_{t+h}(x) = \mathbb{E}_{x(t) \sim \rho_t} \log \rho_{t+h}(x(t + h)), \tag{17}$$

and by the expression (called "instantaneous change-of-variable formula" in normalizing flow literature (Chen et al., 2018), which we derive directly in below)

$$\frac{d}{dt} \log \rho(x(t), t) = -\nabla \cdot \mathbf{f}(x(t), t), \tag{18}$$

we have that for each value of $x(t)$,

$$\log \rho_{t+h}(x(t + h)) = \log \rho(x(t + h), t + h) = \log \rho(x(t), t) - \int_t^{t+h} \nabla \cdot \mathbf{f}(x(s), s)ds.$$

Inserting back to (17), we have

$$\mathbb{E}_{x \sim \rho_{t+h}} \log \rho_{t+h}(x) = \mathbb{E}_{x(t) \sim \rho_t} \log \rho_t(x(t)) - \mathbb{E}_{x(t) \sim \rho_t} \int_t^{t+h} \nabla \cdot \mathbf{f}(x(s), s)ds.$$

The first term is determined by $\rho_t = p_k$, and thus is a constant $c_2$ independent from $\mathbf{f}(x, t)$ on $t \in [kh, (k + 1)h]$. Together with (16), we have shown that

$$\text{r.h.s. of (15)} = c_2 - \mathbb{E}_{x(t) \sim \rho_t} \int_t^{t+h} \nabla \cdot \mathbf{f}(x(s), s)ds) - c_1 + \mathbb{E}_{x(t) \sim \rho_t} V(x(t + h)),$$

which proves (7).

Derivation of (18): by chain rule,

$$\frac{d}{dt} \log \rho(x(t), t) = \frac{\nabla \rho(x(t), t) \cdot \dot{x}(t) + \partial_t \rho(x(t), t)}{\rho(x(t), t)}$$

$$= \left. \frac{\nabla \rho \cdot \mathbf{f} - \nabla \cdot (\rho \mathbf{f})}{\rho} \right|_{(x(t), t)} \quad \text{(by (1) and (2))}$$

$$= -\nabla \cdot \mathbf{f}(x(t), t).$$

$\square$

## B  TECHNICAL DETAILS OF SECTION 4.2

Although the layer-wise training formulation in Section 4.1 enjoys several aforementioned benefits, there exists undesirable movement patterns along the trajectory. Empirically, the movement by initial blocks $f_{\theta_k}$ is much larger than later ones. The blue curve labeled "Phase 1" in Figure A.2a visualizes one typical pattern of the movement measured by $W_2$ distances.

In fact, this phenomenon is not specific to training flow networks by the JKO scheme. It essentially arises due to smaller gradient magnitude at later estimates, which gradually approach a local minimum during optimization. In particular, such irregular movement also appears in gradient descent in vector space. We thus propose a reparametrize-and-refine technique.

*1. Vector-space case*

We first motivate our method with optimization in vector space. Suppose our goal is to find a local minimum $x^*$ of $F(x)$ for a nonlinear differentiable function $F : \mathbb{R}^d \to \mathbb{R}$. Starting at $x^{(0)}$, consider the following sequential optimization problem, where $x^{(t)}$ denotes the estimate at the $t$-th iteration and $h_t$ is a pre-specified regularization parameter:

$$x^{(t+1)} = \arg \min_x F(x) + \frac{1}{2h_t} \|x - x^{(t)}\|_2^2. \tag{19}$$

Using the first order Taylor expansion $F(x) \approx F(x^{(t)}) + \nabla F(x^{(t)})^T (x - x^{(t)})$ at $x^{(t)}$, we get

$$x^{(t+1)} = x^{(t)} - h_t g_t, \ g_t := \nabla_x F(x^{(t)}).$$

Define the *arc length* of iterates $S_t := \|x^{(t+1)} - x^{(t)}\|_2 = h_t \|g_t\|_2$, whereby it appears in practice that the magnitude of $S_t$ is near zero as $x^{(t)} \to x^*$. This issue is typical as a result of small gradient as estimates approach the local minimum. We thus propose Algorithm 2 to resolve this uneven arc length issue, which takes in iterates $x^{(t, \text{old})}$ and step sizes $h_t^{\text{old}}$ from the previous trajectory.

We first motivate and explain the reparametrization step in line 3. Mathematically, we want arc lengths defined using re-optimized values $x^{(t, \text{new})}$ to satisfy $S_t^{\text{new}} \approx \bar{S}$. This is equivalent to requiring $h_t^{\text{new}} \|g_t^{\text{new}}\|_2 \approx \bar{S}$, where $g_t^{\text{new}} := \nabla F(x_t^{\text{new}})$. The quantity $\|g_t^{\text{new}}\|_2$ is unknown before re-optimization takes place, so that we approximate it using $\|g_t^{\text{old}}\|_2 = S_t^{\text{old}} / x^{(t, \text{old})}$. In practice, using the quantity $\bar{S} h_t^{\text{old}} / S_t^{\text{old}}$ alone to update $h_t$ can be undesirable, because larger $h_t^{\text{new}}$ tend to cause non-smooth trajectories and inaccurate final estimates. We thus introduce inertia controlled by parameter $\eta$ and upper bound the largest $h_t$ by $h_{\max}$ to allow more flexibility.

We now explain the refinement step in line 4. We interpolate $C \geq 0$ intermediate points between each pair of $(x^{(t, \text{new})}, x^{(t+1, \text{new})})$. For instance, if $C = 1$, we optimize for the "mid-point" $x^{(t+1/2, \text{new})}$ before reaching $x^{(t+1, \text{new})}$. Using this approach ensures smoother new trajectories $\{x^{(t, \text{new})}\}_{t \geq 1}$ and potentially more accurate final estimate. Figure A.3 illustrates the behavior and our solution on minimizing the Muller-Brown energy potential in $\mathbb{R}^2$.

*2. JKO Flownet reparametrization*

Although Algorithm 2 is developed for re-parametrizing and refining trajectories in vector space $\mathbb{R}^d$, it can be directly used to reparametrize $h$ for JKO-iFlow by replacing the arc length $S_t$ between consecutive iterates in vector space with the $W_2$ movement in probability space of the residual

---

**Algorithm 2** Trajectory improvement (vector-space case)

---

**Require:** Penalty factors $h_t^{\text{old}}$ and iterates $x^{(t,\text{old})}$ for $t = 1, \ldots, T$. Hyper-parameters $h_{\max} > 0$ and $\eta \in (0, 1]$.
1: Compute $S_t^{\text{old}} := \|x^{(t+1,\text{old})} - x^{(t,\text{old})}\|_2$ and $\bar{S} := \sum_{t=1}^{N} S_t^{\text{old}}/T$.
2: **for** $t = 1, \ldots, T'$ **do**
3:     Compute $h_t^{\text{new}} := \min\{h_t^{\text{old}} + \eta(\bar{S}h_t^{\text{old}}/S_t^{\text{old}} - h_t^{\text{old}}), h_{\max}\}$. {▷ Reparametrize}
4:     For $C \geq 1$, store $\hat{h}_t^{\text{new}} := [h_t^{\text{new}}/(C+1), \ldots, h_t^{\text{new}}/(C+1)]$ for $C + 1$ times. {▷ Refine}
5: **end for**
6: Re-optimize (19) for $x^{(t,\text{new})}$ with $\{\hat{h}_t^{\text{new}}\} \cup \infty$.
7: Repeat all steps above until $\text{std}(\{S_t^{\text{new}}\})/\text{mean}(\{S_t^{\text{new}}\})$ is small enough.

---

block. More precisely, let $L$ be the total number of trained blocks via Algorithm 1 and denote $h_k := t_{k+1} - t_k$ as the "step-size" for block $f_{\theta_k}$. Replace the iterates $x^{(t)}$ in vector space with $x(t_k)$, which is the mapping through previous $k - 1$ blocks. Then, the arc length $S_t$ becomes the $W_2$ distance, which can be easily computed using $N$ samples $\{x_i(t_k)\}_{i=1}^N$ along each step of the trajectory. The refinement step thus becomes training additional residual blocks via optimizing (8).

## C  EXPERIMENTAL DETAILS

### C.1  CHOICE OF $t_k$ IN ALGORITHM 1

Recall that to train our JKO-iFlow, one needs as input a sequence of $t_k$, where the $k$-th JKO block integrates from $t_k$ to $t_{k+1}$. Although the selection of $t_k$ varies by problem, we consider two choices in our settings.

- **Constant increment.** Denote $h_k := t_{k+1} - t_k$, We let $h_k \equiv c_1$ for a constant $c_1 > 0$. On many experiments for two-dimensional toy data and high-dimensional data, we use $c_1 = 1$.

- **Constant multiplier.** Given $t_0 > 0$ and a constant $c_2$, we let $t_{k+1} := c_2 t_k$. The rationale is that from empirical evidence, the $W_2$ movement as in (6) tends to be larger at initial blocks than at latter blocks, so that moving later blocks more than the initial ones would enable more uniform movements, thus faciliating the training process. On some experiments for two-dimensional toy data and high-dimensional data, we let $t_0 = 0.75$ and $c_2 = 1.2$.

We acknowledge that many other choices are possible. We also want to emphasize that due to the reparametrization and refinement techniques proposed in Section 4.2, the values of $t_k$ would be adaptively updated based on data, where the adaptive values would yield more uniform $W_2$ movements over blocks as we saw in Section 5.

### C.2  OTHER SETUP DETAILS

All experiments are conducted using PyTorch (Paszke et al., 2019) and PyTorch Geometric (Fey & Lenssen, 2019) . Regarding network architecture

- For simulated 2D data, high-dimensional real data, and MNIST using pre-trained autoencoder: each residual block uses fully-connected layers of the form $d \to H \to H \to d$, where $d$ (resp. $H$) is the feature (resp. hidden nodes') dimension. The hidden dimension vary by example, in the range of 128∼512.

- For conditional graph node feature generation: each residual block uses one Chebnet input layer of order 3 followed by two fully-connected layers. The hidden dimension $H = 64$ in all hidden layers.

The activation function is chosen as Tanh or Softplus with $\beta = 20$. In addition, we use Rugge-Kutta 4 (Lawrence, 1986) to numerically estimate the integrals in the continuous flow model. The Adam optimizer is used with constant learning rate of $1e - 3$ throughout training. Regarding batch sizes, we use 1000 samples for simulated two-dimensional data and 50% of the training samples for the solar graph data. The batch size selections for high-dimensional real data are described in Table A.1.

## C.3 DATASET

For two-dimensional simulated examples, we generate fresh random draws of 10000 training samples at each training epoch. The four high-dimensional real datasets (POWER, GAS, HEP- MASS, MINIBOONE) come from the University of California Irvine (UCI) machine learning data repository. These datasets are commonly used to compare flow models (Grathwohl et al., 2019; Finlay et al., 2020; Onken et al., 2021). The solar dataset as used in iGNN (Xu et al., 2022) is retrieved from the National Solar Radiation Database (NSRDB).

## C.4 MMD METRICS

Besides visual comparison, the maximum mean discrepancy (MMD) (Gretton et al., 2012; Onken et al., 2021) provides a quantitative way to evaluate the performance of generative models. Given samples $\boldsymbol{X} := \{x_i\}_{i=1}^N$ and $\boldsymbol{Y} := \{y_j\}_{j=1}^M$ and a kernel function $k(x,y)$, we compute

$$\text{MMD}(\boldsymbol{X}, \boldsymbol{Y}) := \frac{1}{N^2}\sum_{i=1}^N\sum_{j=1}^N k(x_i, x_j) + \frac{1}{M^2}\sum_{i=1}^M\sum_{j=1}^M k(y_i, y_j) - \frac{2}{NM}\sum_{i=1}^N\sum_{j=1}^M k(x_i, y_j). \quad (20)$$

For our purpose, we use the Gaussian kernel $k(x,y) := \exp\left(-\|x-y\|^2/h\right)$ with bandwidth $h$. We select the bandwidth both as a constant value $h_c = 2$ and via the "sample-median technique" (Gretton et al., 2012) $h_m := 2\text{median}(\{\|x_i - x_j\|^2\}_{i,j})$, where $x_i$ are test samples. We use the same set of 200 test samples to compute $h_m$ in each experiment. We thus denote MMD[c] (resp. MMD[m]) as evaluating (20) using the Gaussian kernel with constant (resp. median) bandwidth, where $\boldsymbol{X}$ (resp. $\boldsymbol{Y}$) denotes true (resp. generated) test sample. Note that a low MMD value indicates two samples $\boldsymbol{X}$ and $\boldsymbol{Y}$ are likely drawn from the same distribution (Gretton et al., 2012). In this setting, MMD is an impartial evaluation metric as it is not used to train JKO-iFlow or any competing methods.

We can also determine the statistical significance of a MMD value. First, compute the threshold

$$\tau := Q_{1-\alpha}(\{\text{MMD}(\boldsymbol{X}[\mathcal{I}_1^b], \boldsymbol{X}[\mathcal{I}_2^b])\}_{b=1}^B), \quad (21)$$

where $Q_{1-\alpha}$ denotes the upper $1 - \alpha$ quantile of a set of scalars and $\mathcal{I}_j^b \subset \{1, \ldots, N\}$ denotes the $j$-th index set at the $b$-th bootstrapping without replacement. Then, under the null hypothesis that $\boldsymbol{X}$ and $\boldsymbol{Y}$ are drawn from the same distribution, this hypothesis is rejected if MMD exceeds the threshold $\tau$. The Type-I error is controlled at level $\alpha$. Thus, if the MMD values by two models both exceed $\tau$, we prefer the model with the smaller MMD. If both values are under $\tau$, then they generate equally well. In our experiments, we use $B = 1000$ bootstraps, each of which has 50% re-sampled test samples.

## C.5 CONDITIONAL GENERATION

We follow the conditional generation scheme as proposed in iGNN (Xu et al., 2022). More precisely, when the response variable $Y$ is a categorical variable taking value in $K$ classes, iGNN designs the target distribution as a Gaussian mixture model. Thus, instead of flowing from data density $p_X$ to noise density $p_Z$, iGNN flows from the conditional data density $p_{X|Y}$ to $p_{H|Y}$, where $H|Y \sim H(\mu_Y, \sigma^2 I)$. One can then minimize the negative log-likelihood $-\log p_{X|Y}$ using $log p_{H|Y}$ and the change-of-variable formula.

To use JKO-iFlow for conditional generation in this setting, we thus only need to modify the objective (8). Instead of using $V_Z$ based on $Z \sim N(0, I_d)$, we would using $V_{H|Y}$ based on the Gaussian mixture $H|Y \sim H(\mu_Y, \sigma^2 I)$.

## C.6 ADDITIONAL RESULTS

We present complete results in addition to those in Section 5. In particular,

- Table A.1 contains the complete numerical results of JKO-iFlow against competitors on high-dimensional real datasets. For ScoreSDE, we use the implementation in (Huang et al., 2021), which computes the evidence lower bound (ELBO) for the data log-likelihood as reported in the last column. In addition, Table A.2 contains MMD and negative log-likelihood results for OT-Flow and FFJORD as taken from the original papers.

Table A.1: Numerical metrics on high-dimensional real datasets, in addition to those Table 2. Comparing to flow-based models, JKO-iFlow takes much less iterations to reach a small enough MMD value. Although ScoreSDE is the fastest, its performance, even under 100 times more iteration than JKO-iFlow, is still worse in terms of MMD[m] on all except GAS. We advocate the comparison using MMD[m] because the results align with visual comparisons in Figure A.1.

| Data Set | Model | # Param | Training | | | | Testing | | |
|---|---|---|---|---|---|---|---|---|---|
| | | | Time (h) | # Iter | Time/Iter (s) | Batch size | MMD[m] | MMD[c] | Neg Loglik |
| **POWER** $d=6$ | | | | | | | $\tau$: **1.75e-4** | $\tau$: **2.84e-4** | |
| | JKO-iFlow | 76K | 0.12 | 0.76K | 0.57 | 10000 | 8.21e-4 | 1.26e-3 | 0.58 |
| | OT-Flow | 76K | 1.21 | 7.58K | 0.57 | 10000 | 5.69e-4 | 9.62e-4 | 0.30 |
| | FFJORD | 76K | 3.40 | 7.58K | 1.61 | 10000 | 1.38e-3 | 1.98e-3 | 0.60 |
| | $W_2$ IResNet | 304K | 0.49 | 7.58K | 0.23 | 10000 | 2.76e-3 | 2.72e-3 | 0.36 |
| | IResNet | 304K | 0.76 | 7.58K | 0.36 | 10000 | 4.50e-3 | 2.49e-2 | 3.37 |
| | ScoreSDE | 76K | 0.08 | 7.58K | 0.04 | 10000 | 1.34e-3 | 6.56e-3 | 3.41 |
| | ScoreSDE | 76K | 0.84 | 75.85K | 0.04 | 10000 | 1.30e-3 | 5.66e-3 | 3.33 |
| **GAS** $d=8$ | | | | | | | $\tau$: **1.93e-4** | $\tau$: **2.83e-4** | |
| | JKO-iFlow | 76K | 0.08 | 0.76K | 0.38 | 5000 | 5.96e-4 | 1.79e-3 | -4.61 |
| | OT-Flow | 76K | 0.72 | 7.60K | 0.34 | 5000 | 1.51e-3 | 3.64e-3 | -4.29 |
| | FFJORD | 76K | 3.49 | 7.60K | 1.65 | 5000 | 3.62e-3 | 6.09e-3 | -2.07 |
| | $W_2$ IResNet | 304K | 0.57 | 7.60K | 0.27 | 5000 | 7.14e-3 | 1.50e-2 | -4.45 |
| | IResNet | 304K | 0.86 | 7.60K | 0.41 | 5000 | 3.26e-3 | 2.72e-2 | -1.17 |
| | ScoreSDE | 76K | 0.04 | 7.60K | 0.02 | 5000 | 1.31e-3 | 1.45e-3 | -3.69 |
| | ScoreSDE | 76K | 0.42 | 76.00K | 0.02 | 5000 | 4.27e-4 | 8.56e-4 | -5.58 |
| **MINIBOONE** $d=43$ | | | | | | | $\tau$: **4.59e-4** | $\tau$: **6.87e-4** | |
| | JKO-iFlow | 112K | 0.03 | 0.32K | 0.33 | 2000 | 7.97e-4 | 1.01e-3 | 13.63 |
| | OT-Flow | 112K | 0.75 | 3.39K | 0.80 | 2000 | 1.23e-3 | 1.01e-3 | 11.93 |
| | FFJORD | 112K | 1.74 | 3.39K | 1.85 | 2000 | 5.47e-3 | 1.04e-3 | 23.45 |
| | $W_2$ IResNet | 448K | 0.80 | 3.25K | 0.89 | 2000 | 1.27e-2 | 1.03e-3 | 16.34 |
| | IResNet | 448K | 1.32 | 3.25K | 1.46 | 2000 | 2.58e-3 | 1.04e-3 | 22.36 |
| | ScoreSDE | 112K | 0.01 | 3.25K | 0.01 | 2000 | 4.29e-3 | 1.03e-3 | 27.38 |
| | ScoreSDE | 112K | 0.09 | 32.48K | 0.01 | 2000 | 4.68e-3 | 1.10e-3 | 20.70 |
| **BSDS300** $d=63$ | | | | | | | $\tau$: **1.35e-4** | $\tau$: **9.63e-5** | |
| | JKO-iFlow | 396K | 0.16 | 1.03K | 0.56 | 5000 | 4.83e-3 | 3.03e-3 | -156.67 |
| | OT-Flow | 396K | 3.50 | 10.29K | 1.22 | 1000 | 8.55e-2 | 8.44e-2 | -142.45 |
| | $W_2$ IResNet | 990K | 2.01 | 10.29K | 0.70 | 1000 | 5.52e-1 | 6.88e-1 | -107.39 |
| | IResNet | 990K | 3.47 | 10.29K | 1.21 | 1000 | 5.42e-1 | 5.95e-1 | -33.11 |
| | ScoreSDE | 396K | 0.01 | 10.29K | 0.005 | 1000 | 5.51e-1 | 6.62e-1 | -7.55 |
| | ScoreSDE | 396K | 0.14 | 102.90K | 0.005 | 1000 | 5.51e-1 | 6.65e-1 | -7.31 |

Table A.2: MMD[c] and negative loglikelihood results of OT-Flow and FFJORD, as taken from (Onken et al., 2021). We include them to compare against ours in Table A.1. The models in previous studies use comparable model size (especially for OT-Flow), where the numerical results in some cases are much smaller than ours due to significantly longer training time.

| Data Set | Model | # Param | Training | | Testing | |
|---|---|---|---|---|---|---|
| | | | Time (h) | # Iter | MMD[c] | Neg Loglik |
| **POWER** $d=6$ | OT-Flow | 18K | 3.1 | 22K | 4.68e-5 | -0.30 |
| | FFJORD | 43K | 68.9 | 29K | 4.34e-5 | -0.37 |
| **GAS** $d=8$ | OT-Flow | 127K | 6.1 | 52K | 2.47e-4 | -9.20 |
| | FFJORD | 279K | 75.4 | 49K | 1.02e-4 | -10.69 |
| **MINIBOONE** $d=43$ | OT-Flow | 78K | 0.8 | 7K | 2.84e-4 | 10.55 |
| | FFJORD | 821K | 9.0 | 16K | 2.84e-4 | 10.57 |
| **BSDS300** $d=63$ | OT-Flow | 297K | 7.1 | 37K | 4.24e-4 | -154.20 |
| | FFJORD | 6.7M | 166.1 | 18K | 6.52e-3 | -133.96 |

- Figure A.1 visualizes the principal component projections of the generated samples by JKO-iFlow and competitors of the high-dimensional real datasets.
- Figure A.2 visualizes components of loss 8 and the resulting generated images.
- Figure A.3 visualizes the trajectory of estimates in $\mathbb{R}^2$ of minimizing the Muller-Brown energy potential.
- Figure A.4 shows additional unconditional and conditional generated samples by JKO-iFlow on toy data.

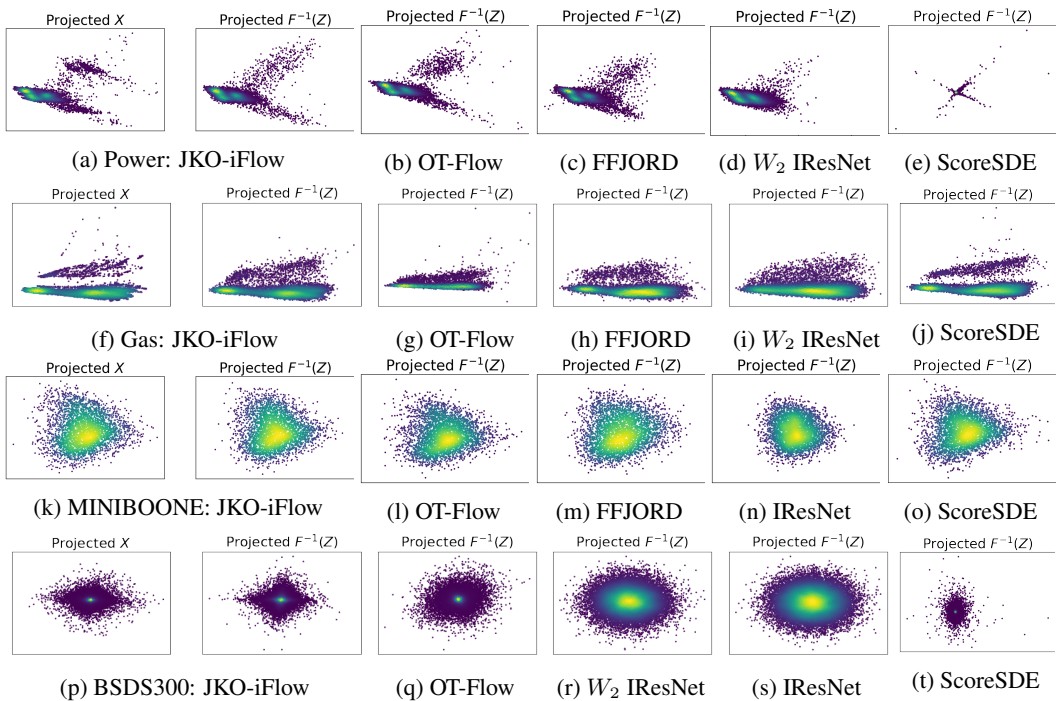

Figure A.1: Generative quality on high-dimensional datasets via PCA projection of generated samples. The generative quality in general aligns with the MMD[m] values shown in Table 2 and A.1.

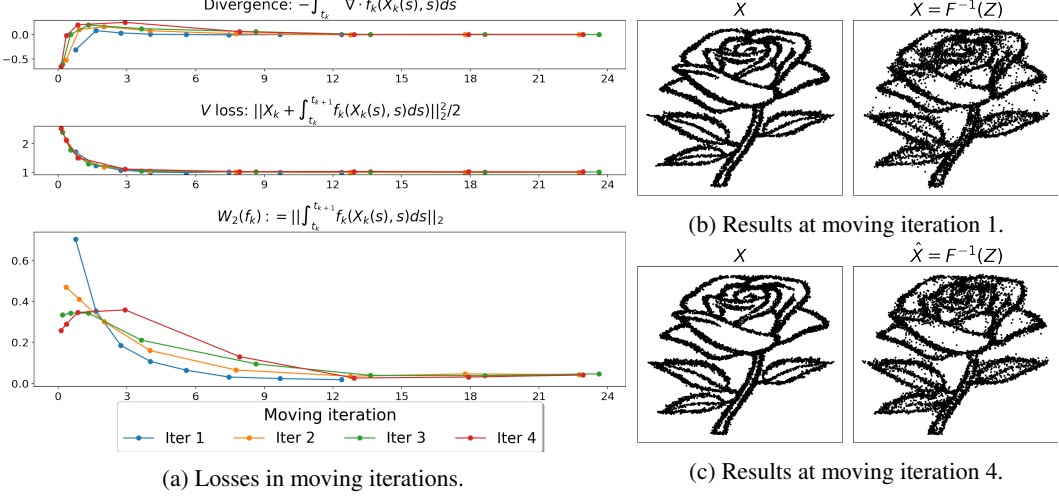

Figure A.2: Rose, reparametrization moving iterations of JKO-iFlow. The plots and setup are identical to Figure 5. We observe improved generative quality after the moving iterations.

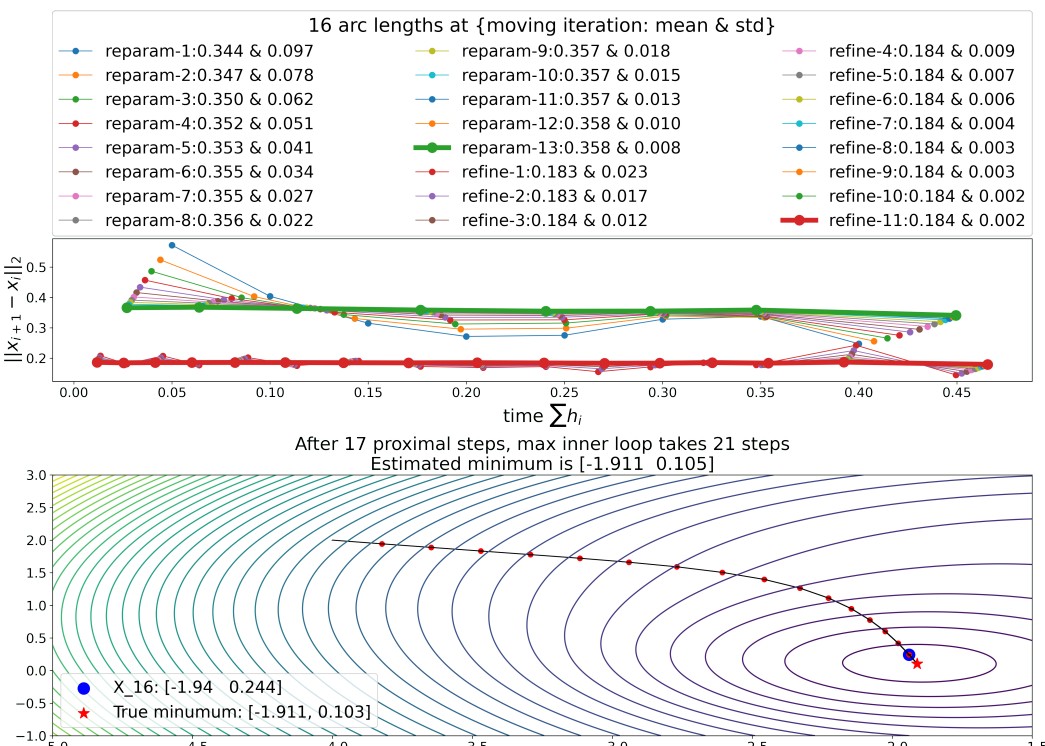

Figure A.3: Reparametrization and refinement moving iterations in vector space based on Algorithm 2. The task is to estimate a local minimizer of the Muller-Brown energy potential. We see that arc lengths between consecutive iterates become more even in magnitude over more reparametrization and refinement moving iterations.

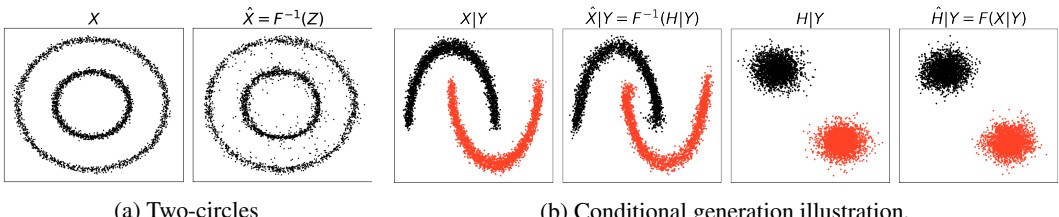

(a) Two-circles

(b) Conditional generation illustration.

Figure A.4: Additional unconditional and conditional generation on simulated toy datasets by JKO-iFlow.

