# OpenReview forum: "Invertible normalizing flow neural networks by JKO scheme"
_ICLR.cc/2023/Conference — Submitted to ICLR 2023_

### Official Review · Reviewer_T8Fn · 2022-10-24

**Confidence:** 3
**Correctness:** 3
**Technical Novelty And Significance:** 3
**Empirical Novelty And Significance:** 2
**Recommendation:** 6

**Clarity, Quality, Novelty And Reproducibility:**

**Questions**
- The objective (8) in the article is actually the $l_2$- regularized (or, $W_2$ - regularized) objective of the original MLE optimization problem for normalizing flows. And this observation reveals some questions regarding the usefulness of the proposed methodology. What you actually do: you separately train $\textit{weak}$ ResNet blocks (weak in the sense that they are weaker than the ResNet network they constitute) with the objective which doesn’t seem to be much more tractable than the original MLE optimization problem. In light of this I have the following questions: At first, if each ResNet block is too powerful to solve the regularized problem can we just use one (or small number of) ResNet blocks to solve the original unregularized problem? In other words, do we really need 74K params (for POWER), 76K params (for GAS), 112K params (for MINIBONE) and 396K params (for BSDS300) of OT-Flow and FFJORD models to achieve the competitive performance (compared to JKO-Flow)? On the other hand, if the ResNet block is too weak to solve the unregularized problem, how to estimate the number of steps and step sizes $h_k$ used at each iteration. Which $h_k$ still makes the regularized problem tractable for the single ResNet block?
- Since we substitute the real ODE with the sequence of JKO problems and use the ResNet block parametrization the real solution and the obtained solution of ODE could diverge. Can we guarantee that the sequence of JKO problems still converges to the target distribution $p_z$?

**Clarity:** The paper is more-or-less well-written.
- The link to the article [2] should be added. Moreover, the authors of [2] called their method “JKO-Flow” which is similar to your approach
- I don’t understand the discussions regarding invertibility of ResNet blocks $f(x, t)$ (section 2.3). We don’t need $f(x, t)$ to be invertible since we model the pushforward transform as $x \rightarrow x + \int\limits_{t_k}^{t_{k + }} f(x(t)) d t$. For the same reason I don’t understand the operation $f := f_k \circ f_{k - 1} \dots \circ f_0$ in the Algorithm 1. The functions $f_i$ don’t correspond to pushforward transform and the composition operations don’t make sense.
- The appendix B (which describes trajectory enhancement) is hard to follow. From my point of view, it can be rewritten in a more pleasant way. (it is just minor comment and it does not affect the recommended score)
- How many ResNet blocks and which $h_k$ were used for high-dimensional tabular data and Mnist experiments?

**Novelty:** The novelty is highlighted in the “strengths” section

**Quality:** The technical quality issues are raised in the “weaknesses” and “questions” sections. Also there are some clarity points-to-fix.

**Reproducibility:** I didn’t run the code provided but it seems to be clearly written and I have no special doubts regarding reproducibility.


**Strength And Weaknesses:**

**Strength**

- To the best of my knowledge, the proposed idea is fresh. It seems that the continuous dynamic parameterization helps to leverage some difficulties faced by previous approaches (like $\log \det J_{T}$ estimation [1] or $\min \max$ objective [2]). Given a good integral estimator the learned JKO-Flow could be easily inverted in a neural ODE manner without need of specific optimization procedures [1].
- The idea to optimize block-composed Neural Network by sort of "divide and conquer" block-wise algorithm also seems to be interesting.

**Weaknesses**
- No comparison with  [2]. The proposed JKO-Flow approach seems to be directly related to their method.
- It would be interesting to see the generative properties of the JKO-Flow approach directly on data space of nontrivial image datasets like CIFAR10 / CelebA. As I understand, the proposed approachcan be straightforwardly adopted for such (image-data) scenarios.


**Summary Of The Paper:**

The paper under consideration proposes a normalizing flow model (called JKO-Flow) which combines JKO scheme and Continuous Normalizing Flows. The previous approaches which dealt with JKO scheme parameterized the pushforward transform $T(x)$  arising in JKO objective either by ICNNs [1, 2] or general NNs of the form $F : \mathbb{R}^D \rightarrow \mathbb{R}^D$ [2] . The main idea of the authors is to replace the pushforward $T(x)$ with continuous dynamics $T(x) = x + \int\limits_{t_k}^{t_{k + 1}} f_k(x(t)) d t$ modeled via the techniques developed for Continuous Normalizing Flows.

[1] Mokrov et. al. Large-Scale Wasserstein gradient flows, https://arxiv.org/pdf/2106.00736.pdf

[2] Fan et. al., Variational Wasserstein gradient flows, https://arxiv.org/pdf/2112.02424.pdf

**Post-rebuttal:** I increase my score by 1

**Summary Of The Review:**

The authors present a new idea which combines CNFs and JKO scheme. However, there are some questions regarding the usefulness of the proposed approach. Moreover, It is recommended to stress-test the method on the image generation problems which are standard benchmarks for many modern Normalizing flows models.

---

> ### Author Response · Authors · 2022-11-19
> **Response to reviewer 4**
>
> Please see the response-to-all comment above for common questions asked by multiple reviewers.
>
> **Q** “how to choose number of steps and step size”
>
> **A** To choose the number of blocks $L$ and the time step $h_k$, we have clarified our algorithm in the revised Section 4. Notably, in the initial trajectory (before reparametrization), the number of blocks is determined by the W2 movement before and after the density transport by the current residual block. This means that the number of blocks used is adaptive to the data - in an extreme case where a single residual block is powerful enough to learn the needed transport such that the transported density $\rho_1$ after training 1 step is already close to $q$, the W2 movement from $\rho_1$ to $\rho_2$ will be small, and then the algorithm will decide to use only 1 Residual block, see the stopping criterion in Algorithm 1.
>
> The adaptive choice of $h_k$ is detailed in Section 4.2, describing the reparametrization of $h_k$ based on the W2 movement of trained residual blocks. The initial choices of $h_k$ are described in Appendix C.1. On the question of the learning in $k$-block of $h_k$ is tractable, first, in theory, when $h_k$ is very small, the solution of the JKO scheme reveals the infinitesimal optimal velocity field (Section 3.2), and it requires the network to express the score function $\log \rho(x,t)$ (but the training objective does not involve learning the score). For finite $h_k$, the optimization is tractable as long as $h_k$ does not exceed a certain threshold. In our reparametrization algorithm (Algorithm 2), we require the adaptive $h_k$ to be less than $h_{\max}$, which is a hyperparameter.
>
> **Q** “needed model size on real data”
>
> **A** As for the needed model size for POWER, GAS, etc. datasets, we have included the baseline from original publications of OT-Flow in Table A.3 (to augment Table 2 in Section 5). The models in previous studies use comparable model sizes, e.g., 78K for MINIBOONE by OT-Flow. Based on this information and our own experiences in experiments, we think the model size reported is the range of model size for flow-based models to give a reasonable performance on these datasets.
>
> **Q** “guarantee the sequence of JKO problems converge to p_z”
>
> **A** Empirically, the stopping criterion in our Algorithm 1 ensures that the transported density by our trained residual network is close to the target distribution. Under generic conditions, as shown in Section 2, we can guarantee that the sequence of JKO problems converges to the target distribution $p_z$, which can be empirically approximated by using a sufficiently large number of residual blocks.
>
> **Q** why resblock blocks should be invertibility and confusing composition notation
>
> **A** The reviewer is correct that $f(x,t)$ does not need to be invertible because it denotes the velocity field for the ODE as in Eqn. (1). In the newly revised statement of Algorithm 1, $f_{\theta_k}$ denotes the trainable residual block, which itself is a neural ODE model on the time interval $[t_k, t_{k+1}]$. We have removed the composition notation to avoid confusion.
>
> **Q** how many ResNet blocks and which h_k are used for high-dimensional tabular data and MNIST experiments
>
> **A** We only use four residual blocks for high-dimensional tabular data. The $h_k$ are initially fixed at 1 (so $t_{k+1}-t_k = 1$), but are trainable via the reparametrization technique proposed in Section 4.2; see Figure 5 for results on MINIBOONE before and after reparametrization.

---

> > ### Comment · Reviewer_T8Fn · 2022-11-23
> > **Response**
> >
> > I thank the authors for the response and for the improvements they made compared to the previous version of the manuscript. Concerning the current version I have the following thoughts:
> > - I agree with reviewer ffZu, that a sort of ablation study which illustrates the profitability of the reparametrization and refinement procedures should be added.
> > - I still don’t understand the discussion regarding the invertibility of Residual block in section 3.3. The residual block doesn’t need to be invertible. Probably the discussion concerns the invertibility of CNF dynamics - in this case the corresponding paragraphs regarding invertibility should be modified to avoid misunderstandings and ambiguities.
> > - From my point of view, the overall flow of the paper can be improved. Do you really need sections 3.3 (and especially 3.2) in the main part of the manuscript (as pointed out by other reviewers, it is known that optimal velocity field is the difference of score functions). Probably, these sections could be partially moved to the appendix.
> >
> > In spite of the mentioned items, I still think that the ideas presented in the paper are interesting and worthy of being presented in the scientific community. That’s why I decided to raise my score by one point. On the other hand, I suppose that the overall quality and flow of the manuscript could be improved.
> >
> > Minor typo:
> > “Rugge-Kutta” -> ‘Runge-Kutta”, last paragraph on page 15.

---

### Official Review · Reviewer_y2DU · 2022-10-24

**Confidence:** 4
**Correctness:** 3
**Technical Novelty And Significance:** 3
**Empirical Novelty And Significance:** 2
**Recommendation:** 5

**Clarity, Quality, Novelty And Reproducibility:**

* The quality of the write-up and presentation can be improved. See weaknesses above for specifics.
* The paper presents a technically interesting idea. However, not all the claims in the introduction have been justified. Furthermore, the quality of the empirical evaluation can be improved.
* The JKO scheme has been explored before but the specific framework proposed in this paper is new.
* The authors have released their code which aids reproducibility. That said, certain experiment setup and metric choices can be better justified.

**Strength And Weaknesses:**

[Strengths]
* The paper presents an interesting method of training a continuous-time normalizing flow to approximate the JKO scheme. This allows the model blocks to be trained sequentially with potentially larger batch sizes.
* While the JKO scheme has been explored before in the machine learning context for large-scale gradient flows (Mokrov et al., 2021), the proposed framework is novel to the best of my knowledge.

[Weaknesses]
* The quality and clarity of the write-up and the overall presentation can be improved.
    * The need for the discussion in Section 2.2 is unclear. The result in Eq. (12) can be directly arrived at by using the fact that JKO scheme recovers the FPK equation which gives us the steepest descent curve. Then a direct matching of the terms in FPK for the OU process with the Liouville Equation gives  the result in Eq. (12). Also see the discussion in Appendix D of Song et al., 2021.
    * Section 3.2 can be better explained. I did not understand what "progressive refinement" is and how exactly does it help the overall training.
    * The scores can be reported in a more readable format. The "e" notation is messy.
    * The paper ends abruptly after the MNIST experiment without a follow-up discussion or conclusion.
    * Minor issues:
        * What is meant by "trajectory obtained by **rest blocks**?"
        * Rephrase "can be **trained by iteslf**".
        * "Th theoretical assumption" -> "The..."
        * "space of probability" -> "space of probability measures".
        * The "T" for termination criterion in Algorithm 1 can be replaced with a different symbol. Currently it coincides with the "T" for the optimal transport map.
        * A few other minor typos here and there. Please proofread.
* On the claims in introduction:
    * The authors discussed non-uniqueness of the flow as a motivation. However, it is unclear how/if the flow induced by the proposed method is unique. How does minimization via Eq. (8) ensure uniqueness? Just empirical evidence is not enough. Previous methods have also provided such evidence.
    * It's unclear if the motivation of pruning unused blocks is actually relevant to other models. This seems to be a issue with the proposed model wherein a naive implementation results in unused blocks.
* On experiments:
    * The experiment results are not convincing enough. Most of the setups considered in the paper are toyish in nature.
    * Why do the MMD results in Table 2 differ significantly from the results reported in Onken et al., 2021? If this is due to a difference in setups, what happens if the setup described in Onken et al., 2021 is used for experiments? What motivated the choice of specific bandwidth in MMD? In my experience, MMD is very sensitive to bandwidth choices. Can a better metric be used?
    * Is it possible to evaluate the log-likelihood under this model? If yes, why did the authors not compare the log-likelihoods of the models?
* On the relation to prior work:
    * With the specific choice of the OU process and the discussion around Eq. (12), the relation to probability flow ODE in Song et al., 2021 should be better discussed.

**Post Rebuttal Edit**: Increased score from 3 to 5.

**Summary Of The Paper:**

The paper proposes a continuous-time normalizing flow model based on the JKO scheme for gradient flows in the Wasserstein-2 space. A neural ODE flow network is trained to approximate the JKO scheme. This results in a block-wise training procedure where each block approximates a JKO step. Several such blocks are trained sequentially and the final flow is obtained via a composition of the blocks. Experiments demonstrate that the proposed method performs competitively against baseline models while being more computationally efficient.

**Summary Of The Review:**

The paper presents a technically interesting idea of training continuous normalizing flows using the JKO scheme. The proposed method results in improved computation efficiency during training. However, this work has issues with the presentation, justification of claims, and quality of empirical evaluation.

---

> ### Author Response · Authors · 2022-11-19
> **Response to reviewer 3**
>
> Please see the response-to-all comment above for common questions asked by multiple reviewers.
>
> **Q** “purpose of section 3.2”
>
> **A** Classical estimate of the JKO scheme gives convergence to the solution to Fokker-Planck (FP) equation in a weak sense, cf. Theorem 5.1 in [Jordan-Kinderleherer-Otto 98] does not give (12). Instead, we derive Eqn (12) using formal expansion of $O(h^k)$ without going into functional analysis details. Meanwhile, Eqn. (12) is not surprising as it “matches” FP with the Liouville equation, as we have commented in Section 3.3 beneath Eqn. (12).
> The purpose of Section 3.2 is to show that objective (8) in the small $h$ limit asks the neural network to approximate the score function $\nabla \log \rho_t(x)$, which in contrast, needs to be obtained via denoising score matching in the diffusion-based methods. This viewpoint is important to explain the relation and distinction between our neural ODE approach to the diffusion-based methods, and we show this by a short deviation. We have revised Section 3.2 with added clarifications and citations.
>
> **Q** “the paper ends abruptly after MNIST”:
>
> **A** Thank you for the suggestion; we have added a discussion section.
>
> **Q** “how the flowed induced by the proposed method is unique”
>
> **A** The Fokker-Planck equation for the OU process is well-posed, and the uniqueness of the solution follows. The discrete-time solution of the JKO scheme was established in (Jordan-Kinderleherer-Otto 98, Proposition 4.1). As proved in Proposition 2.1, the minimization of Eqn (8) is equivalent to that of Eqn. (6) (the transport $T_k$ can be induced by an ODE is by the fact that the Fokker-Planck equation coincides with the Liouville equation under Eqn. (12)), and thus the uniqueness by solving (8) is theoretically guaranteed in the same sense as that of the JKO scheme of solving the FP equation
>
> **Q** Choice of bandwidth in MMD and sensitivity of result to bandwidth
>
> **A** We provide MMD results under the constant bandwidth of 0.5 and the median technique/trick selected bandwidth. The details are described in Appendix C.4. We remark that in [Onken et al., 2021], only the former constant bandwidth is used. We would consider alternative metrics in future analyses.
>
> **Q**  MMD results differ from those in [Onken et al., 2021], cf. Table 2 and Table A.1/A.2
>
> **A** We thank the reviewer for pointing out the difference in the reported baseline in the initial submission. In the revised manuscript, we have included the baseline from the original paper in Table A.2 to make the comparison more complete. Comparing Table A.2 to Table A.1 (our modified experimental setting of the baselines), it can be seen that the original baselines of FFJORD and OT-Flow are trained with a significantly large number of iterations and training time to obtain an indeed better MMD metric.
>
> We modified the setting to focus on performance with less training time. Specifically, we first trained JKO-iFlow until it performed reasonably well on the high-dimensional tabular datasets (i.e., measured by MMD metrics under two bandwidths and the projected PCA visualization). We then trained the alternative approaches using at most ten times more iterations (i.e., mini-batch SGD using Adam) than we did for JKO-iFlow. To make a fair comparison in this setting, we let the number of parameters, batch sizes, optimizer choices, computational platform etc. be the same across different models on each dataset.
>
> As a remark, the model size in  [Onken et al., 2021] is smaller as the network parametrization is continuous over time. The current JKO-iFlow model does not use time-continuity over residual blocks parametrization, which may reduce the model size. We comment on this limitation in the discussion section.
>
> **Q** Report log-likelihood
>
> **A** We have included log-likelihood comparisons in Table A.1.

---

> > ### Comment · Reviewer_y2DU · 2022-11-23
> > **Response**
> >
> > Thanks for you response. Some things have definitely improved in the revision and I have updated my score to reflect that. That said, I agree with the other reviewers that the claims, clarity and experiments still need improvement.

---

### Official Review · Reviewer_tb5k · 2022-10-31

**Confidence:** 5
**Clarity, Quality, Novelty And Reproducibility:** This paper did a bad job in literatur…
**Correctness:** 4
**Technical Novelty And Significance:** 2
**Empirical Novelty And Significance:** 2
**Recommendation:** 3

**Strength And Weaknesses:**

First of all, the paper has a bad definition of notations. For example, the definition of V (is the potential of the equilibrium density) appears in the preliminaries. I think the paper should organize a problem-setting section to make all things clear.

However, in objective (8) there is a V (is the potential of the equilibrium density) included. How to get the V for general distribution? V includes fitting a score, which is very similar to yang's ICLR paper.

The main concern is the realtionship to paper [1,2]. [1] consider the gradient flow of f-divergence and gives out a flow with velocity field f''*nabla (p/p_t). Thrid, how this paper differs from [3]. The only difference is the objective for training but not the gradient flow aiming to approximate. ( If the author claim their objective is better, I need to see evidence)
[1] https://arxiv.org/abs/2012.06094
[3] Johnson R, Zhang T. A framework of composite functional gradient methods for generative adversarial models[J]. IEEE transactions on pattern analysis and machine intelligence, 2019, 43(1): 17-32.

In terms of the literature review, this paper only discusses how they are different in terms of motivation (or how to derive the model), but have no discussion of how the derived model are different from each other.

Missing experiments on CelebA. Missing baseline using diffusion model.

Missing reference:
[1] Liutkus A, Simsekli U, Majewski S, et al. Sliced-Wasserstein flows: Nonparametric generative modeling via optimal transport and diffusions[C]//International Conference on Machine Learning. PMLR, 2019: 4104-4113.
[2] Zhang L, Wang L. Monge-amp\ere flow for generative modeling[J]. arXiv preprint arXiv:1809.10188, 2018.

**Summary Of The Paper:**

This paper proposes using Wasserstein gradient flow of KL dviergence  to construct the flow model.

**Summary Of The Review:**

See above

---

> ### Author Response · Authors · 2022-11-19
> **Response to reviewer 2**
>
>
> Please see the response-to-all comment above for common questions asked by multiple reviewers.
>
> **Q** notation issues and problem-setting section
>
> **A**  The “preliminaries” section (old Section 1.1) has all the needed definitions and notations, which we have separated out as the new Section 2, following the reviewer’s suggestion.
>
> **Q** How to get V for general distribution
>
> **A** We do not attempt to obtain an analytic form of $V$ for general distribution because in most, if not all, related works for normalizing flow and diffusion models we have considered, the equilibrium distribution is always the Gaussian distribution with $V(x)=||x||^2_2$. Thus, this choice of $V$ is already powerful enough for most practical use cases.

---

### Official Review · Reviewer_ffZu · 2022-10-31

**Confidence:** 4
**Correctness:** 3
**Technical Novelty And Significance:** 2
**Empirical Novelty And Significance:** 2
**Recommendation:** 5

**Clarity, Quality, Novelty And Reproducibility:**

The clarity of the paper is okay, although more details and more precise language can further improve the clarity. The source code is given but I did not run it.


**Strength And Weaknesses:**

## Strengths:
* Optimizing the velocity field of the ODE for JKO instead of transport maps that are more common in the existing literature is novel.
* The idea of using one block for each JKO step is interesting and it seems to have resulted in efficiency.


## Weaknesses:
* I found the novelty of the proposed method somewhat limited. The only difference from [Alvarez-Meliset et al. 2021], [Mokrov et al. 2021] is that instead of parameterizing the pushforward map as ICNN (note in either work it's not necessary to use ICNN; arbitrary networks can also be used, following the same reasoning at Lemma A.1), the current work uses neural ODE. I'm not convinced by the superiority of using neural ODE over a pushforward map, which the current work does not compare against. In my opinion, a more severe problem is not addressed: namely at step $k$ all these algorithms need to push initial samples by $k$ steps to obtain samples for the current iteration, which scales overall quadratically in $k$.
* The section 2.2 is not new to my knowledge and references are missing. See a similar derivation in Theorem 3.1 of Liu et al. "Stein Variational Gradient Descent: A General Purpose Bayesian Inference Algorithm".
* I appreciate that two heuristic enhancements are proposed in Section 3.2. However, at first glance, these two enhancements seem contradictory. On the one hand, trajectory reparameterization wants to remove blocks, but on the other hand, progressive refinement requires adding intermediate blocks for large time steps. It would be great to have some ablation studies illustrating the effectiveness of either enhancement.
* Since the target application is generative modeling, and the overall idea follows diffusion models (namely mapping the data distribution to a standard Gaussian, and then inverting the process), I think the authors should compare with diffusion models as well.
* The writing is a bit sloppy overall. There is almost no assumption stated for Lemma A.1 or Proposition 2.1. At the very least we probably need $p$ and $q$ to have a finite second moment along with differentiability assumptions. The proof of 2.1 does not seem novel to me. It is essentially a direct application of the instantaneous change of variable from [Chen et al. 2018] (which is by itself just rewriting the continuity equation). There is missing reference on this formula (see the equation under (17)), and moreover as written this formula is not correct. The derivative needs to be a total derivative, i.e., it should be $d/dt (\log \rho(x(t), t)) = -\nabla \cdot f(x(t), t)$. The sloppiness of the writing is also manifested in the many handwavy sentences in the main text. To give a few examples:
    - Above (4), "Under generic conditions" --- what conditions?
    - In the second paragraph of 2.1, "The solution of (1) ... gives a one-to-one mapping" this should only be true if the time interval is small enough, by Picard–Lindelöf theorem
    - In the last paragraph of page 5, "... it will also have bounded Lipschitz constant". How is this true? There is no assumption on $\rho_t$ having bounded the Lipschitz constant.
    - At the end of page 5, "The analysis is postponed here", where more details could have been given
* For the conditional generation experiments, more details will be helpful. I don't understand what it means by "we evaluate $V$ for a Gaussian mixture $H | Y$".


**Summary Of The Paper:**

This current paper proposes a normalizing flow algorithm that implements the JKO scheme using neural ODE flow blocks. At each time interval, an optimization over a vector field parameterized as ResNet is done to obtain the velocity field that can be integrated to obtain samples at the next step. The normalizing flow is invertible and the objective functional of the JKO scheme is taken to be the KL divergence with respect to a standard Gaussian so that by inverting the flow we can generate the data distribution. The proposed method appears to be more efficient than the alternatives as demonstrated by the experiments.


**Summary Of The Review:**

Overall I think the paper lacks novelty and needs more comparison with other methods (other JKO methods, diffusion models) to demonstrate its effectiveness. The writing can also be improved.

---

> ### Author Response · Authors · 2022-11-19
> **Response to reviewer 1**
>
> Please see the response-to-all comment above for common questions asked by multiple reviewers.
>
> **Q**: “novelty of the proof of 2.1 and missing reference”
>
> **A**: The novelty of our approach lies in using Eqn. (8) as the training objective of a step-wise scheme. As for the so-called instantaneous change-of-variable formula (new Eqn. (18)), we did not include a reference because it directly follows from the definition, and we have added a direct proof therein.
>
> **Q**: Generic conditions above Eqn. (4)
>
> **A**: These conditions are detailed in Bolley et al. (2012), as we have referred to that paper.
>
> **Q**: “The solution of Eqn (1) gives one-to-one mapping”
>
> **A**: The statement holds as long as the ODE is well-posed (including uniqueness of the solution) on the relevant domain of $(t,x)$. We have added a citation to a textbook to clarify this.
>
> **Q**: “.. bounded Lipschitz constant” and Section 3.3
>
> **A**: The smoothness of a function $t$ induces the boundedness of the Lipschitz constant on a compact domain. The smoothness of $\rho_t$ follows from the classical property of the OU process. Section 3.3 is to give intuitive comments to justify the flow's invertibility and how the step-wise scheme makes the learning more tractable. Detailed analysis is not included due to limited space and the scope of the work. We have moved the comment on how to prove model expressiveness to the newly added discussion section.
>
> **Q**: “section 3.2 not new and miss references”
>
> **A**: Thank you for pointing out the missing citation, and we have added the reference to (Liu-Wang 2016). The Thm 3.1 therein gives that the x-transport-map derivative of KL divergence leads to the Stein operator. This, however, does not give the result derived in Section 3.2 (namely optimal ${\bf f}$ equals the difference of score functions) that the JKO scheme considers gradient descent under the Wasserstein-2 metric, which was not considered in (Liu-Wang 2016). We have also added clarification of the purpose of Section 3.2 at the beginning of the section with additional citations.
>
> **Q**: More discussion for conditional generation
>
> **A**: We have included a separate section in Appendix C.5 to explain the details.

---

> > ### Comment · Reviewer_ffZu · 2022-11-23
> > **Response to authors**
> >
> > Thank you for your response. I think the clarity of the paper has improved a lot, and I appreciate the comparison with diffusion models (although it would be more convincing if there are experiments on the image domain where diffusion models excel). Still, I think the novelty of the work is limited (combining JKO and neural ODE). There are many moving parts in the algorithm and it is unclear to me exactly which parts help with improving the numbers compared to prior works. I would like to keep my current score for now.

---

### Author Response · Authors · 2022-11-19
**Response to all reviewers**

We thank the reviewers for the read and valuable feedback, which helps us to improve the paper. We have followed the reviewer’s suggestions to revise the paper, and the major changes include

- We have conducted additional experiments to compare our model with the diffusion model. Specifically, we report the performance of ScoreSDE [Song et al., 2021] on real high-dimensional tabular datasets (Table 2), and the generation of 2D toy examples (Fig. 3).

-  We have expanded the literature review to include additional references and comment on the relationship to our approach.

-  We have revised the presentation of the block-wise training algorithm (Algorithm 1) and the corresponding reparametrization and refinement techniques.

In addition, we have also addressed all the minor places in the updated manuscript.
Please see the detailed response to all reviewers’ comments below. We first respond to questions asked by multiple reviewers in a single place, and we further respond to the rest of the questions to each reviewer respectively.

(R1 = ffZU, R2 = tb5k, R3 = y2DU, R4 = T8Fn.)

1. **Missing references and comparison**

We thank the reviewers for mentioning the related literature. We have added citations to all the mentioned papers and some more, and commented on the relation to our work in the revised manuscript. To respond to the specific questions,

**Q** “novelty and difference from [Alvarez-Meliset et al. 2021], [Mokrov et al. 2021]” (R1)

**A** We have added citations and comments in Section 1.1 to explain the relationship to ICNN-based JKO generating deep network methods and clarify the novelty of our work under the context. In particular, the limitation of ICNN networks has been studied in the literature [Rout et al., 2022; Korotin et al., 2021]. Also, changing the parametrization of the transport map from the gradient of ICNN to a residual network involves additional complications - this was pointed out in detail in the recent work of [Fan et al., 2021], which took both approaches in learning step-wise transport. Given this, we humbly disagree with the reviewer that the framework in [Alvarez-Meliset et al. 2021], [Mokrov et al. 2021] directly allows changing the ICNN parametrization to arbitrary networks. As for the novelty of our neural ODE approach under this background, see more below on the comparison to [Fan et al., 2021], where we answer a question raised by R4.

As for the question of how the complexity of the method scales with $k$, the number of steps, our method scales linearly and in a similar way to previous works using ICNN. For example, Table 2 of [Mokrov et al., 2021] shows that pushing initial densities by $k$ JKO steps is linear in $k$ in terms of the number of function evaluations of the pushforward map.

**Q**  “using neural ODE/multiple residual blocks v.s. learning a single pushforward map” (R1/R4)

**A** Flow-based invertible generative models (e.g., iResNet, FFJORD, etc.) use consecutively several ResNet blocks to model the transport of data samples from data distribution $p = p_X$ to the target (normal) distribution $q = p_Z$ so that the generation of $X \sim  p_X$ can be conducted by sampling $Z \sim p_Z$ and flow “backward” from $q$ to $p$. (The forward and backward direction may be reserved in certain models, but as long as the network is invertible, it is equivalent). An alternative approach, as correctly pointed out by R1, is to find a one-step transport from p to q, e.g., the optimal transport by the Monge-Ampere potential [Zhang et al., 2018] based on OT theory. The two approaches may have advantages in different scenarios - note that finding the “one-step” optimal transport itself is a challenging problem, especially for high dimensional data. The many-step flow-based method conceptually breaks down the hard problem of flowing towards q into many steps of the potentially easier problem of flowing from $\rho_k$ to $\rho_{k+1}$ which is closer to $q$, and the JKO scheme provides a principled way to find this sequence of $\rho_k$ which is theoretically guaranteed to approach $q$ after a certain number of steps. As has been pointed out in Section 3.3, the advantage lies in solving a more tractable learning problem in each step, both in an approximation theory point of view and in training the network.

---

> ### Author Response · Authors · 2022-11-19
> **Response to all reviewers (cont.)**
>
> **Q** “relationship to paper [Liutkus et al., 2019] [Zhang et al., 2018] (R2）
>
> **A** The training objective of [Liutkus et al., 2019] adopts the sliced-Wasserstein distance, which is only an approximation to the Wasserstein metric, in training the generative model. Our loss objective in Eq. (8) solves for the JKO step, which tracks the Wasserstein gradient flow in the probability space.
> The formulation in [Zhang et al., 2018] is based on the Monge-Ampere flow, which solves for a single transport map rather than composed transport from multiple steps; the latter is taken in our work. Please see the question “using neural ODE/multiple residual blocks v.s. learning a single pushforward map” above for comparing these two approaches in learning normalizing flow models. Method-wise, we propose block-wise training following the JKO scheme, which differs from the approach of [Zhang et al., 2018].
>
> **Q** “relationship to  [Johnson et al., 2019] (R2)
>
> **A** [Johnson et al., 2019] aims to improve GAN without using the minimax formulation, and the framework differs from the normalizing flow models to which our work belongs. In particular, our neural-ODE approach for the normalizing flow model trains an invertible network and allows density estimation, uncertainty quantification, etc., which may not be obtained directly from GAN models.
>
> **Q** “comparison to [Song et al., 2021]” (R2/R3)
>
> **A** Our model trains an invertible neural ODE normalizing flow network, which differs from diffusion models (neural SDE models) such as [Song et al., 2021] in several aspects. We have included detailed comments in Section 1.1. In particular, our approach does not involve denoising score matching to learn the score function from sampled data. In addition, our neural ODE model is invertible, while the diffusion models are not. Empirically, we have added experimental comparison to  [Song et al., 2021] in Section 5.
>
> **Q** “comparison with [Fan et al., 2022]” (R4)
>
> **A** We have expanded the literature review to include a comparison with [Fan et al., 2022]. In particular, the variational approach in  [Fan et al., 2022]  requires a $D$-net-like inner-loop training, which is not needed in our approach. Also, our neural ODE model trains an invertible flow network which allows computation of push-forward distribution, resolving a question raised by [Fan et al., 2022]. Please see more details in Section 1.1.
>
> **References:**
> - [Alvarez-Meliset et al. 2021] Alvarez-Meliset et al., Optimizing functionals on the space of probabilities with input convex neural networks, https://openreview.net/pdf?id=dpOYN7o8Jm 2021
> - [Mokrov et. al. 2021] Mokrov et. al., Large-Scale Wasserstein gradient flows, https://arxiv.org/pdf/2106.00736.pdf 2021
> - [Liutkus et al., 2019] Liutkus A, Simsekli U, Majewski S, et al. Sliced-Wasserstein flows: Nonparametric generative modeling via optimal transport and diffusions[C]//International Conference on Machine Learning. PMLR, 2019: 4104- 4113. https://arxiv.org/abs/1806.08141
> - [Zhang et al., 2018] Zhang L, Wang L. Monge-amp\ere flow for generative modeling[J]. arXiv preprint arXiv:1809.10188, 2018., https://arxiv.org/abs/1809.10188
> - [Johnson et al., 2019] Johnson R, Zhang T. A framework of composite functional gradient methods for generative adversarial models[J]. IEEE transactions on pattern analysis and machine intelligence, 2019, 43(1): 17-32., https://ieeexplore.ieee.org/document/8744312
> - [Song el. al. 2021] Song el. al., Score-Based Generative Modeling through Stochastic Differential Equations, ICLR 2021
> - [Fan et al. 2022] Fan et. al., Variational Wasserstein gradient flows, https://arxiv.org/pdf/2112.02424.pdf, 2022
> - [Rout et al., 2022] Litu Rout, Alexander Korotin, and Evgeny Burnaev. Generative modeling with optimal transport maps. In International Conference on Learning Representations, 2022.
> - [Korotin et al., 2021] Alexander Korotin, Vage Egiazarian, Arip Asadulaev, Alexander Safin, and Evgeny Burnaev. Wasserstein-2 generative networks. In International Conference on Learning Representations, 2021.

---

> > ### Author Response · Authors · 2022-11-19
> > **Response to all reviewers (cont.)**
> >
> > 2. **Experiments on other image dataset** (R2,R3,R4)
> >
> > Reviewers are correct that the proposed approach can be directly applied to such image data scenarios, and we appreciate the suggestion. Due to limited time and computing resources during the rebuttal period, we could not finish larger image experiments, which involved more expensive training in composing convolutional layers. Such experiments are left as future work.
> >
> > 3. **Comparisons with diffusion model** (R1,R2)
> >
> > We have added the baseline of [Song et al., 2021] (SocreSDE) on toy & high-dimensional tabular data, where our JKO-iFlow yields competitive or better results.
> >
> > 4. **Motivation and presentation of the section on trajectory reparametrization and refinement** (R1,R3,R4)
> >
> > [Motivation of progressive refinement R3] The motivation for introducing progressive refinement is to adaptively train more residual blocks so that the flow obtained by the trained residual blocks is smoother and more regular.
> >
> > [Presentation and block removal] We have clarified the algorithm in Section 4.2. Specifically, the “removal” of blocks is not needed in Algorithm 1 (training of the trajectory before reparametrization), and a stopping criterion can determine the number of blocks. A typical choice is the amount of W2 movement. We have also revised the presentation of numerical results and the details of Algorithm 2 in the appendix to clarify this point.

---

### Decision · Program_Chairs · 2023-01-20

**Decision:**

Reject

**Justification For Why Not Higher Score:**

Three reviewers agree to reject the manuscript while one reviewer weakly accepts the paper after reading the authors rebuttal. Therefore following the majority opinion, the decision is made.

**Justification For Why Not Lower Score:**

N/A

**Metareview: Summary, Strengths And Weaknesses:**

This manuscript proposes a normalizing flow model (called JKO-Flow) which combines JKO scheme and Continuous Normalizing Flows that implements the JKO scheme using neural ODE flow blocks. At each time interval, an optimization over a vector field parameterized as ResNet is done to obtain the velocity field that can be integrated to obtain samples at the next step. The normalizing flow is invertible and the objective functional of the JKO scheme is taken to be the KL divergence with respect to a standard Gaussian so that by inverting the flow we can generate the data distribution. The proposed method shows improved computational efficiency during training. Despite that the manuscript presents a technically interesting idea of training continuous normalizing flows using the JKO scheme, worthy of being presented in the scientific community, the current version has issues with the presentation, justification of claims, and quality of empirical evaluation raised by the reviewers. The overall quality and flow of the current manuscript need improved to be ready for publication.